# Interferon induced circRNAs escape herpesvirus host shutoff and suppress lytic infection

Sarah E Dremel[1,5], Takanobu Tagawa [1,5], Vishal N Koparde [2,3], Carmen Hernandez-Perez[1], Jesse H Arbuckle [4], Thomas M Kristie [4], Laurie T Krug[1] & Joseph M Ziegelbauer [1✉]

## Abstract

To globally profile circRNAs, we employ RNA-Sequencing paired with chimeric junction analysis for alpha-, beta-, and gamma-herpesvirus infection. We find circRNAs are, as a population, resistant to host shutoff. We validate this observation using ectopic expression assays of human and murine herpesvirus endoribonucleases. During lytic infection, four circRNAs are commonly induced across all subfamilies of human herpesviruses, suggesting a shared mechanism of regulation. We test one such mechanism, namely how interferon-stimulation influences circRNA expression. 67 circRNAs are upregulated by either interferon-β or -γ treatment, with half of these also upregulated during lytic infection. Using gain and loss of function studies we find an interferon-stimulated circRNA, circRELL1, inhibits lytic Herpes Simplex Virus-1 infection. We previously reported circRELL1 inhibits lytic Kaposi sarcoma-associated herpesvirus infection, suggesting a pan-herpesvirus antiviral activity. We propose a two-pronged model in which interferon-stimulated genes may encode both mRNA and circRNA with antiviral activity. This is critical in cases of host shutoff, such as alpha- and gamma-herpesvirus infection, where the mRNA products are degraded but circRNAs escape.

**Keywords** Circular RNAs; Herpesviruses; Host Shutoff; Interferon-stimulated Genes
**Subject Categories** Immunology; Microbiology, Virology & Host Pathogen Interaction; RNA Biology

## Introduction

Herpesviridae is a family of large, double-stranded DNA viruses with a biphasic life cycle, a lytic (replicative) and latent (quiescent, immune evasive) phase. There are nine species known to infect humans, including the alpha-herpesvirus Herpes Simplex Virus-1 (HSV-1), beta-herpesvirus human cytomegalovirus (HCMV), and gamma-herpesvirus Kaposi sarcoma-associated herpesvirus (KSHV). Herpesviruses are a major public health concern with individuals testing seropositive for at least three of the nine species by adulthood (Ablashi et al, 1995; Baillargeon et al, 2000; Bate et al, 2010; Bradley et al, 2013; Dowd et al, 2013; Zhang et al, 2022). Infection is asymptomatic for many individuals but, in cases of immune-compromise—such as transplant recipients, neonates, and those with HIV/AIDS—these viruses have devastating effects. HSV-1 commonly causes recurrent oral and genital lesions, but can also cause herpes keratitis, herpetic whitlow, and encephalitis (Gopinath et al, 2023). HCMV is the most common congenital infection in addition to a severe opportunistic infection in transplant recipients and individuals with HIV/AIDS (Griffiths and Reeves, 2021). KSHV is the etiological agents of several cancers including Kaposi sarcoma and primary effusion lymphoma (Cesarman et al, 2019). Murine gamma-herpesvirus 68 (MHV68) has close genetic homology to KSHV and serves as a tractable animal model for pathogenesis (Dong et al, 2017; Wang et al, 2021). To date only varicella zoster virus has an FDA-approved vaccine. In addition, we lack antivirals capable of targeting the latent reservoir and there is no therapeutic agent capable of clearing these viruses.

Viruses evolve unique mechanisms to invade hosts, alter cellular pathways, and redirect cellular factors for viral processes. In parallel, the host employs a barrage of proteins and RNA species to combat infection. An emerging class of transcripts, circular RNAs (circRNA), has recently been implicated in this host-pathogen arms race (Chen et al, 2017; Harper et al, 2022; Li et al, 2017; Tagawa et al, 2023; Yao et al, 2021a). CircRNAs are single-stranded RNAs circularized by 5' to 3′ covalent linkages called back-splice junctions (BSJs). High-throughput sequencing paired with chimeric transcript analysis enables global circRNA detection and quantification (Szabo and Salzman, 2016). These techniques find circRNAs to be ubiquitously expressed in an array of organisms and tissues (Maass et al, 2017; Salzman et al, 2013). CircRNAs are also expressed by viruses, including KSHV, Epstein Barr Virus (EBV), human papillomavirus, Merkel cell polyomavirus, hepatitis B virus, and respiratory syncytial virus (Abere et al, 2020; Tagawa et al, 2018; Toptan et al, 2018; Ungerleider et al, 2018; Yao et al, 2021b; Zhao et al, 2019; Zhou et al, 2018). The mechanism underlying host circRNA synthesis, back-splicing, is catalyzed by the spliceosome and regulated by RNA binding proteins (RBPs) and tandem repeat

[1]HIV and AIDS Malignancy Branch, National Cancer Institute, Bethesda 20892, USA. [2]CCR Collaborative Bioinformatics Resource, National Cancer Institute, Bethesda 20892, USA. [3]Frederick National Laboratory for Cancer Research Advanced Biomedical Computational Sciences, Leidos Biomedical Research, Inc., Frederick 21701, USA. [4]Laboratory of Viral Diseases, National Institute of Allergy and Infectious Diseases, Bethesda 20892, USA. [5]These authors contributed equally: Sarah E Dremel, Takanobu Tagawa. ✉E-mail: ziegelbauerjm@nih.gov

elements which mediate interaction of BSJ flanking sequences (Ashwal-Fluss et al, 2014; Conn et al, 2015; Jeck et al, 2013; Starke et al, 2015). CircRNAs function as miRNA sponges, protein scaffolds, and transcriptional enhancers (Kristensen et al, 2019). CircRNAs are generally classified as noncoding RNAs (ncRNAs), although they possess the capacity for cap-independent translation (Chen and Sarnow, 1995; Legnini et al, 2017; Pamudurti et al, 2017). Recently, we identified a host circRNA, circRELL1, that increased the growth of KSHV-infected cells while suppressing the lytic cycle, thereby promoting the viral latency program (Tagawa et al, 2023). Additional host circRNAs modulate viral infection (circHIPK3-KSHV, circPSD3-Hepatitis C virus) and are implicated in virus-driven tumorigenesis (circARFGEF1-KSHV, circNBEA-Hepatitis B virus) (Chen et al, 2020; Harper et al, 2022; Huang et al, 2017; Yao et al, 2021a). Furthermore, circRNAs made by spliceosome-independent mechanisms leads to activation of the pattern recognition receptor (PRR), RIG-I (Chen et al, 2017). Another report found that circRNAs, as a class, sequester the RBP encoded by interleukin enhancer-binding factor 3 (NF90/NF110) and this axis modulates vesicular stomatitis virus infection (Li et al, 2017).

As circRNAs lack ends, they are generally resistant to exoribonucleases with approximately 2.5-fold longer half-lives than their linear counterparts (Enuka et al, 2016; Memczak et al, 2013). CircRNAs are also more stable in the extracellular space, a feature which has led to much interest in their potential use as a diagnostic biomarker (Wang and Liu, 2022). Circularity, however, does not prevent susceptibility of circRNAs to endoribonucleases (endoRNases) such as RNase L and RNase P (Liu et al, 2019; Park et al, 2019). Herpesviruses also express endoRNases, e.g., HSV-1 virion host shutoff (vhs), EBV BamHI fragment G leftward open reading frame 5 (BGLF5), KSHV shutoff and exonuclease (SOX), and MHV68 murine SOX (muSOX). These viral proteins drive a phenomenon called "host shutoff", which, in part, ablates the immune response by degrading interferon-stimulated genes (Abernathy and Glaunsinger, 2015; Smiley, 2004). The viral endoRNases display broad nucleolytic activity in vitro relying on viral and host protein adapters in vivo to fine tune their RNA substrates (Daly et al, 2020). These adapters facilitate a preference for translationally competent RNA leaving ncRNA enriched in the escapee population (Abernathy et al, 2014; Covarrubias et al, 2011; Feng et al, 2005; Gaglia et al, 2012; Page and Read, 2010; Shiflett and Read, 2013). Circularity itself provides some protection from vhs cleavage in vitro, however circRNAs containing an internal ribosome entry site can still be targeted (Shiflett and Read, 2013). In the context of HSV-1 infection, a recent study reported enrichment of circRNAs relative to their colinear gene products, which was not observed in the context of a vhs-null virus (Friedl et al, 2022).

To define host circRNAs commonly regulated by herpesviruses, we performed comparative circRNA expression profiling of cells infected with alpha- (HSV-1), beta- (HCMV), and gamma-herpesviruses (KSHV; MHV68, a murine model of KSHV). We profiled cell culture and animal models, spanning lytic and latent infection. During lytic HSV-1, KSHV, and MHV68 infection, circRNAs were, as a population, unaffected by host shutoff. Ectopic expression assays with human and murine herpesvirus endoRNases confirmed this observation. This agrees with prior reports regarding HSV-1 vhs-mediated decay (Friedl et al, 2022; Shiflett and Read, 2013) and expands the observation to gamma-herpesvirus endoRNases. We identified four

human and twelve murine circRNAs commonly upregulated after infection across subfamilies of herpesviruses. The most upregulated pathways in our models of HSV-1, HCMV, and KSHV lytic infection were related to immunity. Thus, we examined if circRNA expression was affected by treatment with various immune stimuli (LPS, poly I:C, CpG) or type I and II interferons (IFN). 67 circRNAs were upregulated by IFN treatment, with half of these also upregulated during viral infection. Finally, we tested if one of these interferon-stimulated circRNAs, circRELL1, echoed the antiviral function of its colinear gene product. Using gain and loss of function studies, circRELL1 was found to inhibit lytic HSV-1 infection. These results echo our prior finding, that circRELL1 inhibits lytic KSHV infection (Tagawa et al, 2023), and hints at a common mechanism of action that spans disparate cell types (fibroblast vs. endothelial) and viruses (alpha vs. gamma-herpesviruses). Our data suggests this class of host shutoff escapees may have largely unprobed potential as immunologic effectors.

## Results

### CircRNA profiling of alpha-, beta-, and gamma-herpesvirus infection

As alternative splicing products, circRNAs share almost complete sequence identity with their linear counterparts derived from the same gene. We used CIRCExplorer3-CLEAR to quantify the unique sequence of circRNAs, namely the 5′ to 3′ back splice junctions (Ma et al, 2019). CIRCExplorer3 also calculates CIRC-score, the number of reads spanning circRNA BSJs (circ fragments per billion mapped bases, circFPB) against reads spanning mRNA forward splice junctions (linear fragments per billion mapped bases, linearFPB). We profiled RNA-Seq data from alpha-, beta-, and gamma-herpesvirus infection, in cell culture and animal models (Figs. 1A, EV1A and EV2A). The data was a combination of our own RNA-Seq data (HSV-1, KSHV, MHV68) in addition to a previously published dataset (HCMV) (Data ref: Oberstein and Shenk, 2017). For all RNA-Seq, excluding the HCMV dataset, ERCC (External RNA Controls Consortium) spike-ins were used to control for global transcriptomic shifts caused by infection. We have summarized all the circRNA and gene expression profiling as interactive data tables (Dataset EV1–6) with their utility demonstrated in Fig. EV3.

In primary lytic infections we identified 151, 70, and 316 upregulated (log$_2$ fold change (log$_2$FC) Infected/Uninfected >0.5) human circRNAs for HSV-1, HCMV, and KSHV, respectively (Fig. 1B,C, Dataset EV7). Four circRNAs were upregulated across viruses (Fig. 1B). A similar number of circRNAs were downregulated after infection (log$_2$FC < 0.5), with six overlapping between models (Fig. 1B). In HSV-1 infection, disparate circRNA/mRNA expression changes were particularly evident, resulting in dramatic CIRCscore shifts (Fig. 1C). Multiple loci had CIRCscores >5, indicating the circRNA, rather than the colinear mRNA, was the predominant mature transcript for that gene. An increasing CIRCscore with infection was also present for other lytic infection models (Figs. EV2C and EV1C). We extended our analysis to mouse models, including HSV-1 infected trigeminal ganglia and MHV68 infected cell lines. We identified 113 murine circRNAs upregulated by HSV-1 infection, although none overlapped between latency, explant-induced reactivation, and drug-enhanced reactivation (Fig. EV1B,C, Dataset EV7).

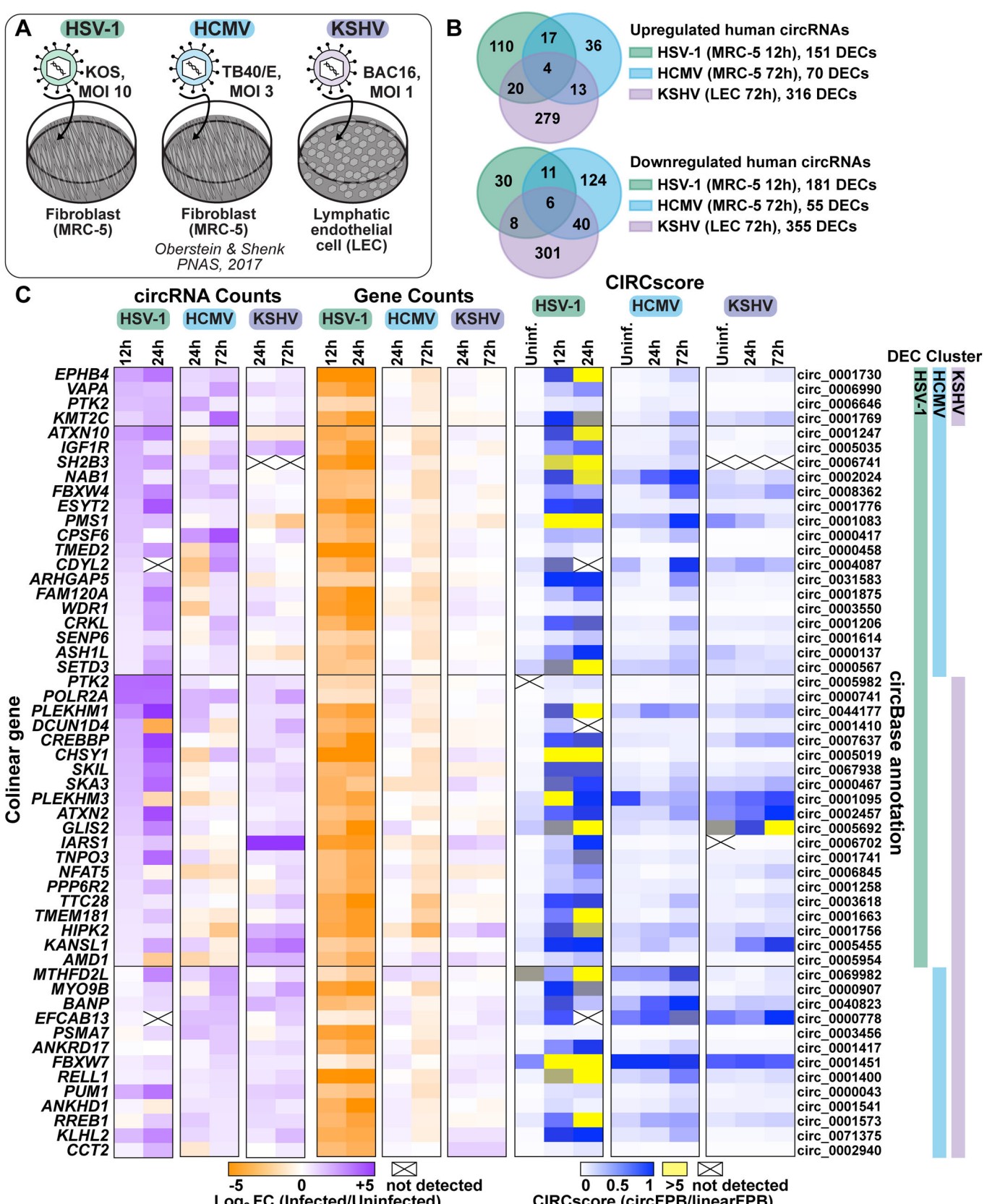

**Figure 1.  Human circRNAs upregulated in de novo lytic infection models.**

(A) Infographic for infection models used in this study. (B) Overlap of differentially expressed circRNAs (DECs) detected by bulk RNA-Seq from HSV-1 ($n = 2$–4), HCMV ($n = 2$), and KSHV ($n = 2$) infection. (C) Heatmaps for DECs which overlap between viruses, with DEC clusters indicating which model the circRNA was found to be significantly upregulated within. Data is plotted as circRNA counts (log$_2$FC), Gene counts (log$_2$FC), or CIRCscore (circFPB (fragments per billion mapped bases)/linearFPB). Data Information: Bulk RNA-Seq data was normalized to ERCC spike-in controls. Log$_2$FC is relative to a paired uninfected control. Differential expression $p$-values were calculated using RankProd non-parametric permutation tests. DECs were those with raw back splice junction (BSJ) count across the sample set >10, log$_2$FC > 0.5 or <−0.5, and $p$-value < 0.05. Heatmap values are the average of biological replicates.

There were 72 murine circRNAs upregulated by MHV68 infection, with four overlapping between primary infection and lytic reactivation (Fig. EV2B,C, Dataset EV7). There were 12 circRNAs in common between HSV-1 and MHV68 infection models, of these circMed13l (mmu_circ_0001396) was upregulated in all infection models (Figs. EV1 and EV2).

Differentially expressed circRNAs (DECs) common across disparate virus and cell models hints at a common mode of induction. To investigate this, we performed overrepresentation analysis (ORA) on the colinear genes of circRNAs expressed in human infection models (Appendix Fig. S1). ORA identified enrichment of genes involved in cellular senescence for all fibroblast (MRC-5) models, likely representing that infection is performed in G0 cells. Interestingly, genes within the lysine degradation pathway were enriched after infection of HSV-1, HCMV, and KSHV (Appendix Fig. S1). One biological function of circRNA is regulation of mRNA expression through miRNA sponging (Kristensen et al, 2019). We performed circRNA-miRNA-mRNA network analysis for circRNA commonly upregulated by herpesvirus infection (circEPHB4, circVAPA, circPTK2, and circKMT2C) (Appendix Fig. S2). In silico analysis predicted miRNA-mRNA interaction nodes which were enriched for mRNA involved in adaptive immunity including MHC complex assembly, TAP complex binding, and peptide antigen stabilization, suggesting a potential role in antiviral immunity for these commonly upregulated circRNAs.

## Global distribution shifts for mRNA, lncRNA, and circRNA during lytic infection

In HSV-1 infection of MRC-5 (Fig. 1C), circRNA upregulation was at odds with the stark decrease in colinear gene expression. A similar trend was visible, albeit less notable, for other lytic infection models (Figs. EV2C and EV1C). This finding led us to question if circRNAs were resistant to the global downregulation of host RNAs which occurs during lytic infection. Using ERCC normalized RNA-Seq datasets we plotted read distribution shifts for HSV-1, KSHV, and MHV68 lytic infection (Fig. 2). We compared expression changes for protein-coding genes (mRNA), long noncoding RNAs (lncRNAs), and circRNAs. In Fig. 2A we observed a sharp decrease in host mRNA levels, with a median log$_2$FC of −4.3 (HSV-1 12 hpi), −2.2 (KSHV 72 hpi), and −1.9 (MHV68 18 hpi). This coincides with high levels of viral gene expression. As has been previously reported (Abernathy et al, 2014; Covarrubias et al, 2011; Gaglia et al, 2012), this effect was partially ablated for lncRNAs with median log$_2$FC of −3.9 (HSV-1 12 hpi), −1.3 (KSHV 72 hpi), and −1.5 (MHV68 18 hpi) (Fig. 2B). Strikingly, host circRNAs were globally resistant to HSV-1 shutoff and instead exhibited a general upregulation (log$_2$FC + 1.3) by 24 hpi (Fig. 2C). During KSHV and MHV68 infection, host circRNA expression changes were tri-modal with a downregulated, unaffected, and upregulated subpopulation. However, the bulk of circRNA species were again

largely unchanged with median log$_2$FC of +0.2 (KSHV 72 hpi) and −0.2 (MHV68 18 hpi). We examined if the tri-modal distribution of host circRNAs during KSHV reactivation may be explained by sequence differences. A degenerate motif "UGAAG" can increase substrate recognition for the KSHV endoRNase, SOX (Gaglia et al, 2015). We posited that underrepresentation of this motif in circRNAs may allow them to escape KSHV host shut off. We performed motif scanning (Table 1) for our tri-modal circRNA populations to assess incidence of the motif. Proportions of motif-harboring circRNAs did not correlate with up- or down-regulation upon reactivation. Overall density of degenerate motifs was, however, highest in downregulated circRNAs and lowest in upregulated circRNAs, suggesting a potential role of UGAAG frequency in SOX resistance. Our analysis demonstrates that host circRNAs, as a species of RNAs, are resistant to the global downregulation of RNA which occurs during lytic herpesvirus infection.

## CircRNAs are resistant to viral endonuclease-mediated decay

Figure 2 plots expression changes across an entire RNA class. To investigate if similar trends occurred for circRNAs and mRNAs derived from the same gene, we evaluated expression shifts for circRNAs (circFPB) and mRNAs (linearFPB) as log$_2$FC (Infected/Uninfected) for HSV-1, KSHV, and MHV68 lytic infection (Fig. 3A). Each dot is a gene that can be alternatively spliced, generating both circRNAs and mRNAs. Echoing our results in Fig. 1C and Fig. 2A,C, circRNA abundance increased while mRNA decreased for a vast majority of genes after HSV-1 infection (Fig. 3A). For KSHV and MHV68, mRNA downregulation was consistently more pronounced than circRNA downregulation. Bulk RNA-Seq examines steady-state transcript abundance, averaging the effects of transcriptional activity in addition to co- and post-transcriptional processing such as splicing and decay. To examine what most influences circRNA upregulation we examined our previously published nascent RNA-Seq and ChIP-Seq data (Data ref: Dremel and DeLuca 2019; Data ref: Dremel et al, 2022a; Data ref: Dremel et al, 2022b; Data ref: Dremel et al, 2022c) for a subset of genes colinear to circRNAs upregulated during HSV-1 infection (Appendix Fig. S3). All four genes (POLR2A, EPHB4, CREBBP, PLEKHM1) had a drop in RNA Polymerase II (Pol II) and TATA-binding protein (TBP) occupancy by 4 hpi, consistent with published mechanisms of host transcriptional shutoff during HSV-1 infection (Abrisch et al, 2015; Birkenheuer et al, 2018; Dremel and DeLuca, 2019; McSwiggen et al, 2019). In the case of EPHB4, CREBBP, and PLEKHM1 we also observed a drop in 4 thiouridine (4sU)-Seq read coverage, consistent with decreased nascent transcripts. This data suggests circRNA expression changes—for at least this subset—are related to co- or post-transcriptional processing.

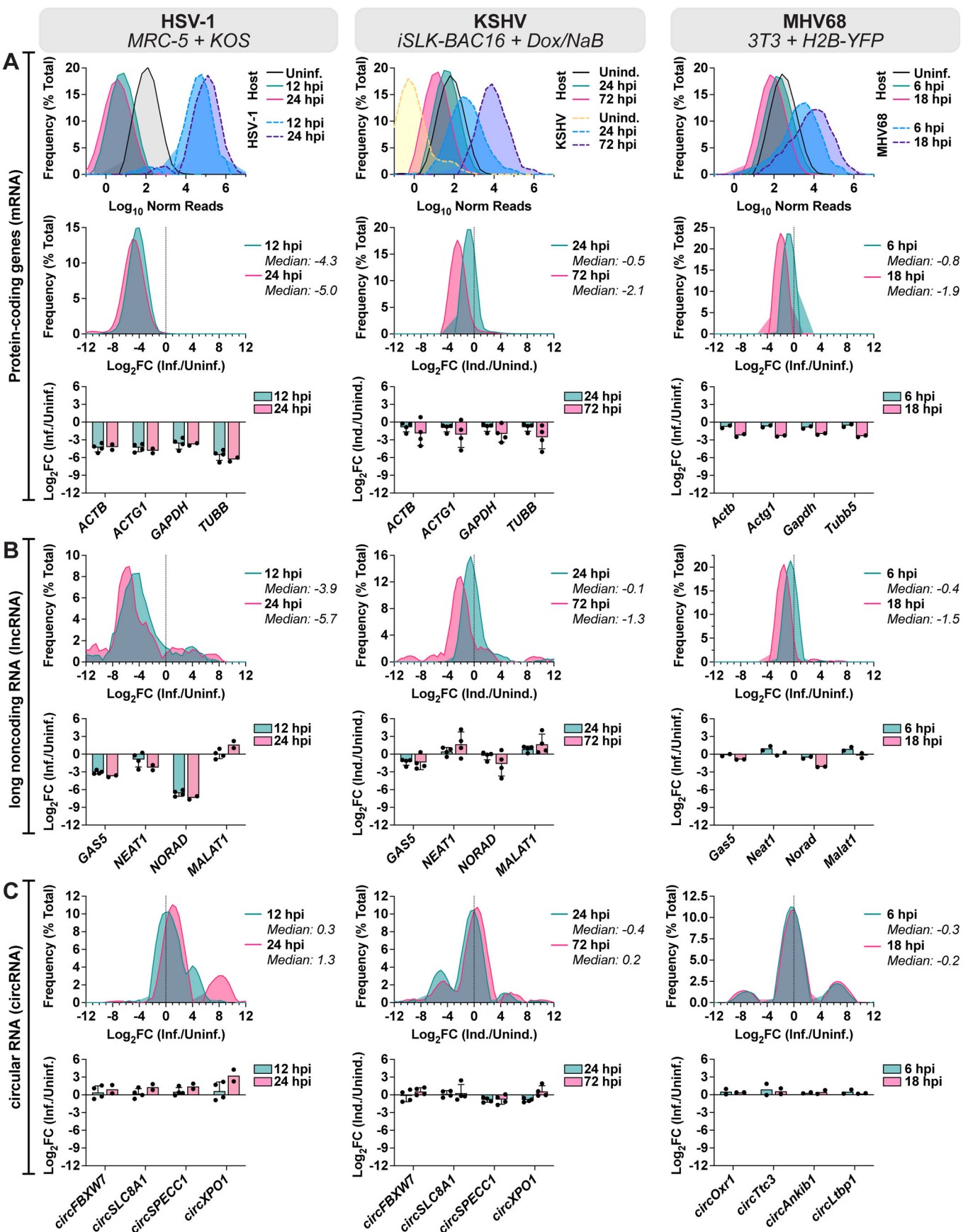

◀ **Figure 2. Global distribution shifts for mRNA, lncRNA, and circRNA during lytic infection.**

(A–C) Bulk RNA-Seq data from HSV-1 infection (MRC-5 infected with KOS MOI 10, $n = 2$–4), KSHV lytic reactivation (iSLK-BAC16 induced with 1 μg/mL doxycycline 1 mM sodium butyrate, $n = 4$), and MHV68 infection (3T3 infected with H2B-YFP MOI 5, $n = 2$). (A) Data for protein-coding genes plotted as relative frequency distribution for $\log_{10}$ ERCC normalized reads or $\log_2$FC (limited to top 10,000 most highly expressed genes). $\log_2$FC for representative protein-coding genes is plotted in column bar graphs. (B) Data for lncRNA genes plotted as relative frequency distribution for $\log_{10}$ normalized reads or $\log_2$FC (limited to top 100 most highly expressed genes). $\log_2$FC for representative lncRNA genes is plotted in column bar graphs. (C) Data for circRNA plotted as relative frequency distribution for $\log_{10}$ normalized BSJ reads or $\log_2$FC (limited to top 100 most highly expressed circRNAs). $\log_2$FC for representative circRNAs is plotted in column bar graphs. Data Information: Bulk RNA-Seq data was normalized to ERCC spike-in controls, $\log_2$FC is relative to a paired uninfected or uninduced control. Frequency distribution plots are the average of biological replicates. In column bar graphs, data points are biological replicates, column bars are the average, and error bars are standard deviation.

**Table 1. Analysis of SOX substrate-targeting motif.**

| Subset: Log₂FC (3 dpi/ Uninduced) | # circRNA | # circRNA in circAtlas | # UGAAG motifs | # Motifs per circRNA | # Motifs per 1000 nt | UGAAG+ circRNA | |
|---|---|---|---|---|---|---|---|
| | | | | | | # circRNA | % Total |
| Upregulated: ≥0.5 | 42 | 39 circRNA (22,459 nt) | 71 | 1.9 | 3.2 | 37 | 88% |
| Unchanged: <0.5 and >−0.5 | 28 | 28 circRNA (20,975 nt) | 95 | 3.7 | 4.5 | 26 | 93% |
| Downregulated: ≤−0.5 | 30 | 25 circRNA (12,900 nt) | 104 | 4.7 | 8.1 | 22 | 73% |

The presence of SOX substrate-targeting motifs "UGAAG" was determined for subsets of human circRNAs from our KSHV reactivation model (iSLK-BAC16).

Given this observation, we explored if differences in viral endoRNase-mediated decay might explain the disparate expression profiles for circRNA and their colinear genes. Using the approach of Rodriguez et al (2019), plasmids expressing HSV-1 vhs, EBV BGLF5, KSHV SOX, and MHV68 muSOX were transfected into 293T cells for 24 h. Ectopic gene expression and Thy1.1 surface expression were confirmed by RT-qPCR and flow cytometry, respectively (Appendix Fig. S4). Transcript levels were quantified relative to a paired GFP vector control (Fig. 3B). As expected, transfection with viral endoRNases decreased *GAPDH* expression (Rodriguez et al, 2019). Conversely, ncRNAs such as *U6 snRNA*, *7SL*, and *MALAT1* were either unaffected or were increased. We also recapitulated prior findings, regarding escape of the *SHFL* mRNA from SOX-mediated decay (Rodriguez et al, 2019). We then tested expression changes of circRNAs and colinear mRNAs using divergent or convergent primers, respectively. Across all genes tested and viral endoRNases transfected, circRNA were more resistant to decay as compared to their colinear gene product. CircRNA and colinear mRNA expression differed most significantly in vhs and SOX transfected 293T (Fig. 3B). Thus, we propose circRNAs as a general class of host shutoff escapees. Our data agrees with prior work from HSV-1 (Friedl et al, 2022), arguing circRNA resistance to viral endoRNases is primarily responsible for the disparate expression changes of circRNAs relative to their colinear mRNA gene products.

## Detection of interferon-stimulated circRNAs (ISCs)

The most significantly upregulated pathways during HSV-1, HCMV, and KSHV infection fell largely within the category of immune responses, with IFN-β and -γ predicted as upstream regulators (Appendix Fig. S5). This led us to investigate if circRNA expression may be modulated by innate immune signaling. We treated fibroblast (MRC-5), lymphatic endothelial cell (LEC), and B-cell lymphomas (Akata-, BJAB, Daudi) with immune stimulants including lipopolysaccharide (LPS), CpG DNA, poly I:C, IFN-β and -γ (Fig. EV4). A canonical interferon-stimulated gene (ISG), *ISG15*, was measured as a surrogate for immune stimulation. We measured

circRELL1 expression, as this circRNA was upregulated in HCMV, KSHV, and to a lesser extent, HSV-1 infection (Fig. 1C). circRELL1 was upregulated in LEC treated with LPS or IFN-γ and B-cell lymphomas treated with poly I:C, LPS, or IFN-β (Fig. EV4). *ISG15* activation did not always correlate with circRELL1 expression—notably CpG and poly I:C treatment largely failed to induce expression. In addition, IFN-β and -γ caused inverse phenotypes in circRELL1 expression when comparing LEC and B-cell models (Fig. EV4B). These findings demonstrate that circRELL1 can be upregulated by Toll-like receptor engagement and type I and II interferon stimulation, and the expression profile varies by cell type and mechanism of immune stimulation.

To globally profile interferon-stimulated circRNAs (ISCs) we performed RNA-Seq on fibroblast, lymphatic endothelial, and B-cells treated with IFN-β and -γ. B-cells (Akata) were only treated with IFN-β as we found them refractory to IFN-γ (Fig. EV4B). Transcriptomic analysis identified strong upregulation of many canonical ISGs including *OAS1*, *OAS2*, *OASL*, *IFIT1*, *IFITM1*, *MZ1*, *HLA-DRB1*, *HLA-DQA1*, and *HLA-DMA* after IFN treatment (Fig. 4A, Dataset EV6). The extent and range of ISGs was most pronounced for B-cells treated with IFN-β ($n = 4173$ DEGs) (Fig. 4A,B) and smallest for fibroblast treated with IFN-β ($n = 472$ DEGs). We identified approximately a dozen interferon-stimulated circRNAs in each model (Fig. 4C). Of these, circEPSTI1 (hsa_circ_0000479) was upregulated in all conditions (Fig. 4D). We tested if circRELL1 and circEPSTI1 expression changes could be recapitulated in peripheral blood mononuclear cells (PBMCs) and observed a 1.5 and 9-fold increase, respectively, after IFN-β treatment (Appendix Fig. S6).

A number of colinear mRNAs and circRNAs were stimulated by interferon treatment, including transcripts derived from *CRIM1*, *EPSTI1*, *ZCCHC2*, *SP100*, *AFF1*, *B2M*, and *WARS* (Fig. 4B,D). The typical mechanism of ISG induction relies upon promoter activation by factors like IRF3, IRF7, IRF9, STAT1, and STAT2. If this mechanism is similarly responsible for circRNA upregulation, we would expect circRNA levels to correlate with changes in their colinear gene products. We plotted gene expression relationship for ISCs and colinear transcripts (Fig. EV5). A subset of ISC-hosting genes, *EPSTI1*, *SP100*, *B2M*, and *WARS*, had a direct relationship between circRNA

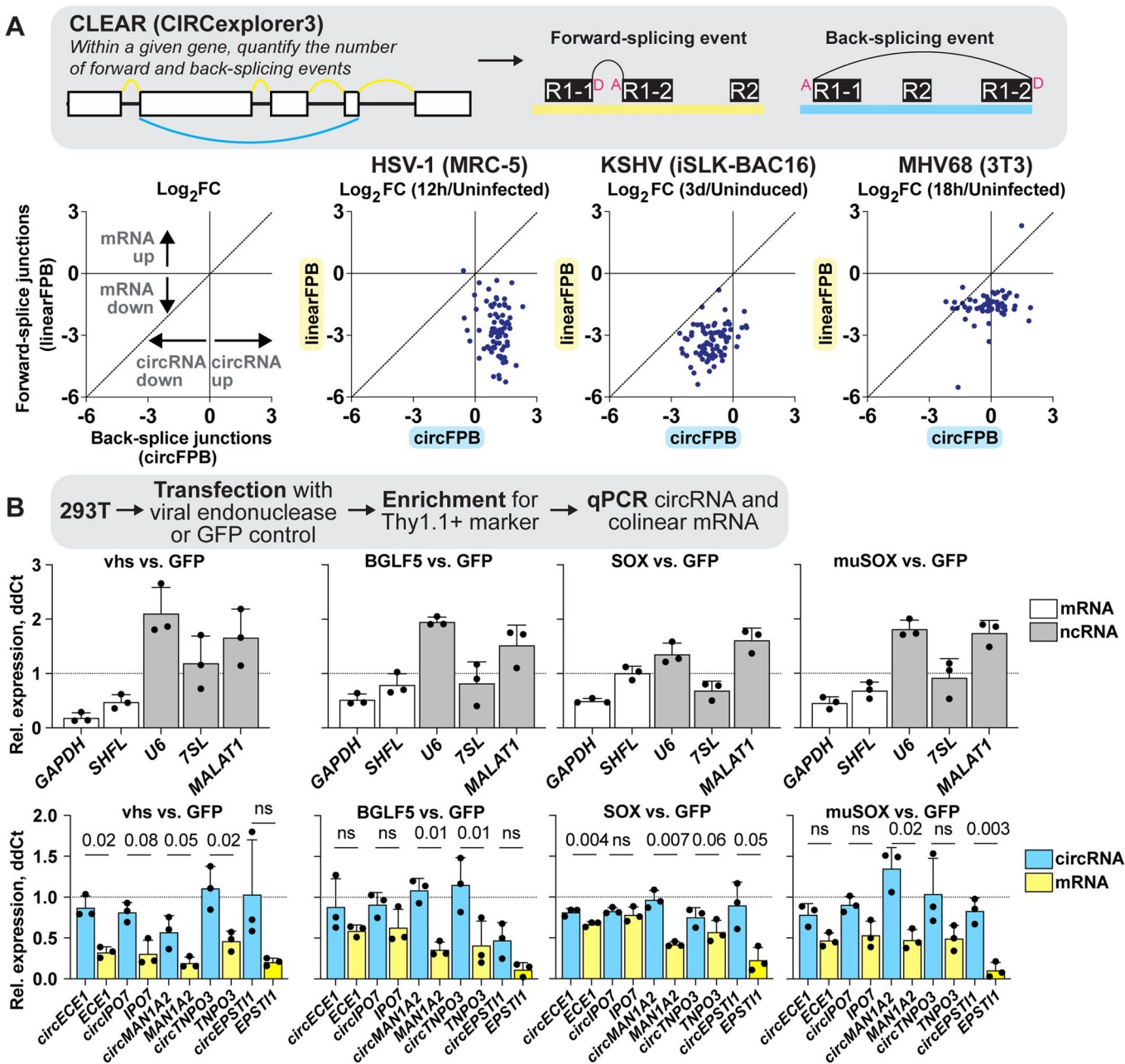

**Figure 3. CircRNA are resistant to viral endonuclease-mediated decay.**

(A) Bulk RNA-Seq data from HSV-1 (MRC-5 infected with KOS MOI 10 for 12 h, $n = 4$), KSHV (iSLK-BAC16 reactivated with Dox/NaB for 3 days, $n = 4$), and MHV68 (3T3 infected with H2B-YFP MOI 5 for 18 h, $n = 2$) infection. Graphs are limited to genes where raw BSJ and forward splice junction (FSJ) counts were >1 across all biological replicates. Each dot is the average $\log_2$FC of biological replicates, with each dot representing linearFPB and circFPB for a distinct gene. (B) Ectopic expression assays of viral endonucleases (vhs, BGLF5, SOX, muSOX) versus control (GFP) vector in 293T cells. RNA was collected after 24 h and transcripts quantified by qPCR ($n = 3$). Data is plotted as relative expression (ddCt) using 18S rRNA as the reference gene, and relative to a paired GFP transfected sample. Data points are biological replicates, column bars are the average, and error bars are standard deviation. Data Information: Paired t-tests were performed comparing circRNA and colinear mRNA expression, $p$-values < 0.1 are labeled and not significant (n.s.) indicates $p$-values ≥ 0.1. Source data are available online for this figure.

and colinear gene expression changes. However, this was not globally the case, with linear regression $R^2$ values ranging from <0.3 (MRC-5 + IFN-β, LEC + IFN-γ, Akata + IFN-β) to ≥ 0.6 (MRC-5 + IFN-γ, LEC + IFN-β) (Fig. EV5A). We next examined overlaps between our interferon-stimulated and infection-stimulated circRNAs

(Fig. 1, Dataset EV1, EV2, EV3, and EV6). EBV-stimulated circRNAs (Fig. 4E) were from our previously published microarray dataset which compared EBV-positive versus EBV-negative Akata cells (Data ref: Tagawa et al, 2023). Approximately half of the ISCs detected were also upregulated during models of HSV-1, HCMV, KSHV, or EBV

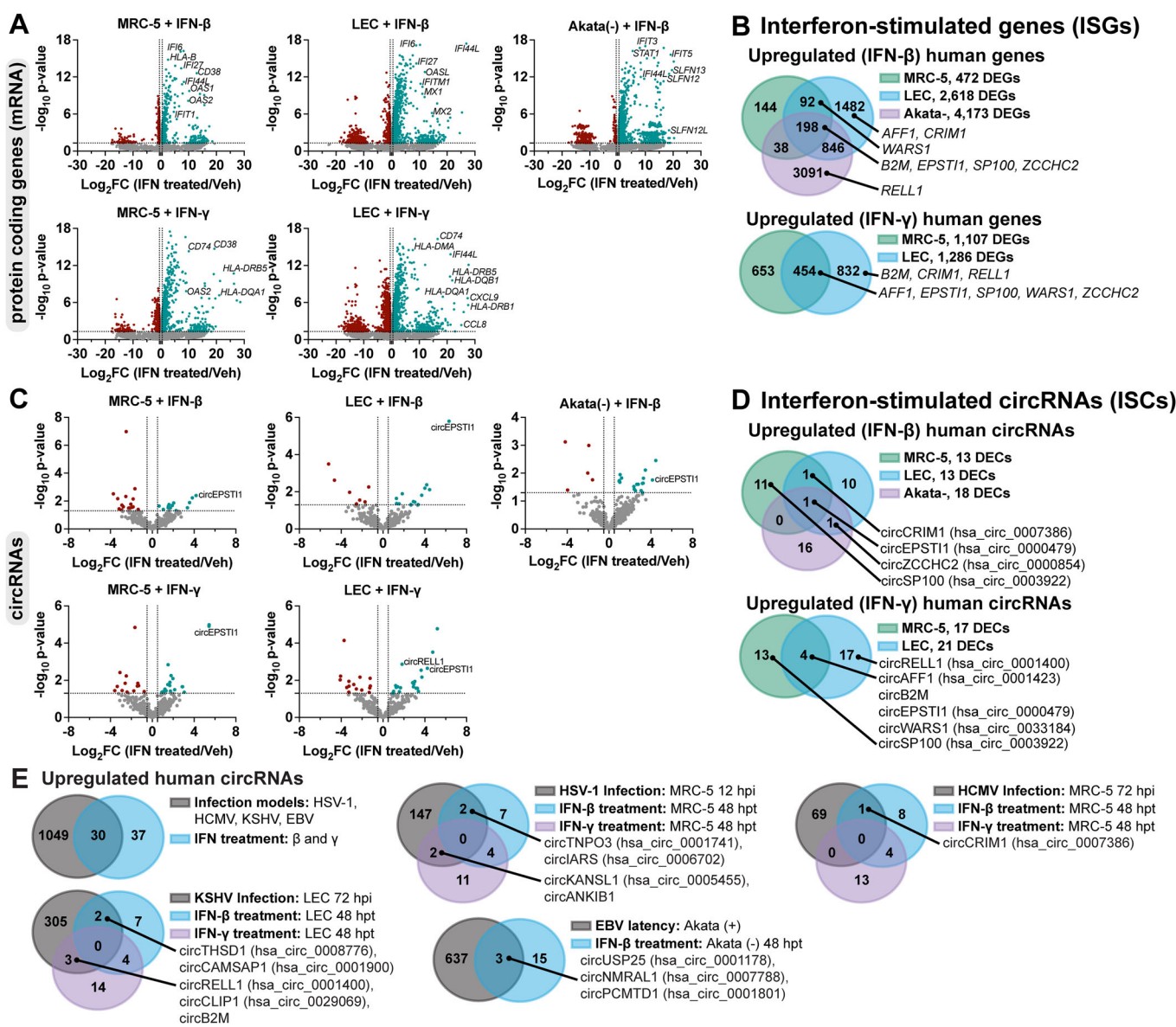

**Figure 4. Detection of interferon-stimulated circRNAs (ISCs).**

(A–E) Bulk RNA-Seq data from MRC-5, LEC, or Akata- cells were treated with recombinant interferons for 48 h ($n = 3$). MRC-5 and LEC were treated with IFN-β and -γ (25 ng/mL conc). Akata- were treated with IFN-β (10 ng/mL). (A,C) Volcano plots for normalized protein-coding gene (mRNA) or circRNA reads. (B,D) Venn diagrams of significantly upregulated circRNA or mRNAs. (E) Overlap of circRNAs upregulated during herpesvirus infection (Fig. 1) or interferon-stimulation. Data Information: Bulk RNA-Seq data was normalized to ERCC spike-in controls. Log$_2$FC was calculated relative to a paired untreated sample. Adjusted p-values (Benjamini and Hochberg) were calculated using EdgeR quasi-likelihood F test method. Differentially expressed genes or circRNAs were those with raw counts across the sample set >10, log$_2$FC > 0.5, and p-value < 0.05.

infection (Fig. 4E). These findings demonstrate tunability in circRNA expression with dependence on cell type and immune stimulation.

## circRELL1 restricts HSV-1 lytic infection

In this study, we found circRELL1 to be induced by HSV-1 (1.3-fold), HCMV (1.4-fold), and KSHV (2-fold) infection (Fig. 1C). circRELL1 was also upregulated by immune stimulants including LPS, poly I:C, IFN-β and -γ (Fig. EV4, Appendix Fig. S6). Our lab has previously reported inhibition of lytic KSHV infection by

circRELL1 (Tagawa et al, 2023). Thus, we questioned if it could be broadly antiviral, by perturbing circRELL1 within the context of HSV-1 infection. To test the impact of loss of function, we depleted circRELL1 48 h prior to infection using siRNAs targeting the BSJ. Our siRNAs were specific to the circRNA species with the colinear mRNA, *RELL1*, unaffected by siRNA treatment (Appendix Fig. S7). MRC-5 cells were infected at a low (0.1 plaque forming units (PFU)/cell) and high (10 PFU/cell) multiplicity of infection (MOI). We achieved significant depletion (4- to 10-fold decrease) of circRELL1 as compared to a Non-Targeting Control (Fig. 5A,D).

    

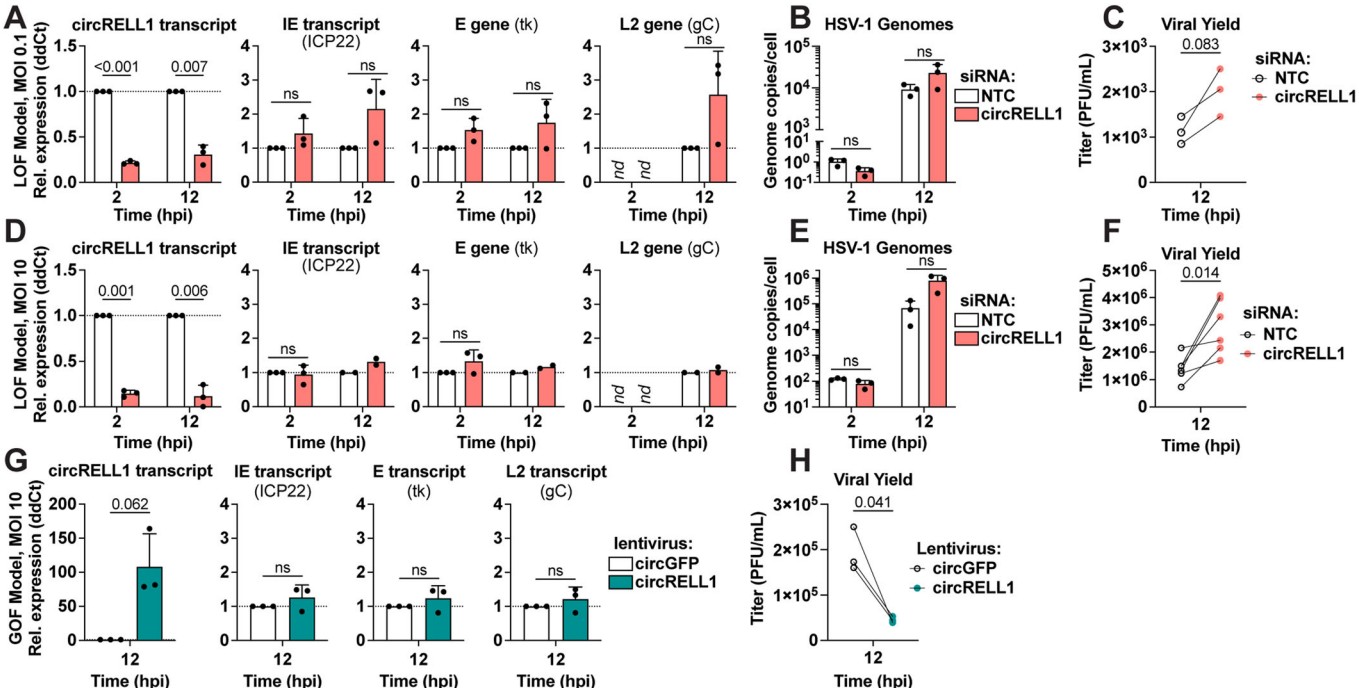

**Figure 5. circRELL1 (hsa_circ_0001400) restricts HSV-1 lytic infection.**

(A–F) circRELL1 was depleted in MRC-5 cells using siRNAs for 48 h and subsequently infected with HSV-1 strain KOS at MOI of 0.1 or 10 PFU/cell. Data is relative to a paired Non-Targeting Control siRNA (NTC). (G,H) MRC-5 cells were infected with a lentivirus expressing circRELL1 for 48 h and subsequently infected with HSV-1 strain KOS at MOI of 10 for 12 h. Data is relative to a control lentivirus expressing circGFP. (A,D,G) RNA was collected from the cell fraction and reverse transcribed. qPCR data is plotted as relative expression (ddCt) using 18 S rRNA as the reference gene (n = 2–3). (B,E) DNA was isolated from the cell fraction and assessed by qPCR for viral and host genome copies (n = 3). (C,F,H) Supernatant was collected at 12 hpi and assessed by plaque assay. Data points (C,H n = 3; F n = 6) are biological replicates, column bars are the average, and error bars are standard deviation. Data Information: Paired two-tailed t-tests were performed, p-values < 0.1 are labeled and not significant (n.s.) indicates p-values ≥ 0.1. Source data are available online for this figure.

There was no significant change in viral gene expression for immediate early (IE), early (E), and true late (L2) transcripts after circRELL1 knockdown. Viral entry was similarly unaffected with the level of viral genomes at 2 hpi comparable between Non-Targeting Control and siRNA against circRELL1 (Fig. 5B,E). The effect of circRELL1 depletion was most apparent when measuring infectious viral yield, with a 1.8- and 2.2-fold increase for low and high MOI infections, respectively (Fig. 5C,F). For gain of function studies, we transduced circRELL1 in MRC-5 with a replication-null lentivirus for 48 h followed by infection with HSV-1 at high MOI (10 PFU/cell). By 60 h post-lentivirus infection, circRELL1 expression was 100-fold greater than our lentivirus control that harbors circGFP (Fig. 5G). While we observed no alterations in IE, E, or L2 viral transcripts, viral yield was decreased 4.4-fold by circRELL1 overexpression (Fig. 5G,H). Our loss and gain of function models agree, supporting an anti-lytic role for circRELL1 in HSV-1 infection that appeared to be independent of viral gene expression.

## Discussion

CircRNAs are gaining traction as important factors at the virus-host interface. CircRNAs have an interesting combination of physical features: versatility to interact with DNA, RNA, and proteins simultaneously, longer half-lives than colinear mRNAs, and secretion to extracellular spaces. Some of these features are shared by other molecules like lncRNAs or microRNAs, but circRNAs uniquely possess all those flexible and multi-faceted characteristics. In this study, we probed the role of circRNAs during herpesvirus infection. We performed comparative circRNA expression profiling and identified four host circRNAs commonly upregulated by all subfamilies of human herpesviruses (Fig. 1). Whether there are universal functions of these commonly regulated circRNAs is not clear but overrepresentation analysis of colinear transcripts showed their enrichment in cell division/senescence pathways and lysine degradation (Appendix Fig. S1). In silico circRNA-miRNA-mRNA interaction networks predict circEPHB4, circVAPA, circPTK2, and circKMT2C may regulate cellular immunity via repression of miRNA-mediated decay of mRNAs involved in antigen presentation (Appendix Fig. S2). We identified a subset of circRNAs induced by herpesvirus infection and interferon stimulation (Fig. 4). Since circRNAs and mRNAs expressed from the same loci can have distinct targets and functions, they can be thought of as polycistronic genes. We propose a two-pronged model in which interferon-stimulated genes may encode both mRNA and circRNA with immune-regulatory

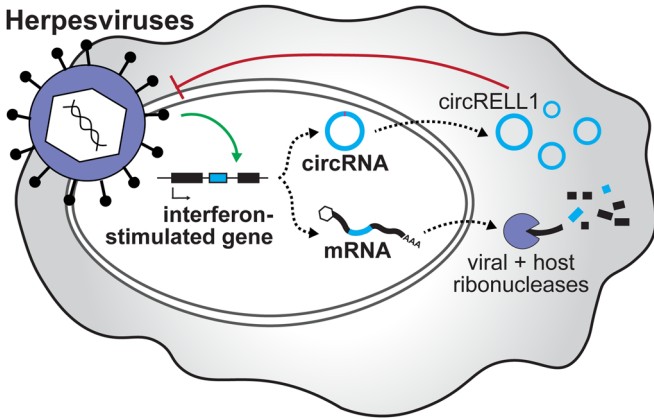

**Figure 6. Proposed polycistronic model for interferon-stimulated genes.**

We propose a polycistronic model in which interferon-stimulated genes can produce both mRNA and circRNA with antiviral activity. This is critical in cases of host shut off, such as alpha- and gamma-herpesvirus infection, where the mRNA product is degraded but circRNA escapes. The interferon-stimulated circRNA, circRELL1, exemplifies this model. EBV, KSHV, and HCMV infection upregulates circRELL1 expression which functions to suppress lytic infection of HSV-1 and KSHV.

activity (Fig. 6). This is critical in cases of host shutoff, such as alpha- and gamma-herpesvirus infection, where the mRNA product is degraded but circRNA escapes.

CircRNAs were resistant to ectopic expression of HSV-1, EBV, KSHV, and MHV68 endoribonucleases (Fig. 3B), suggesting them to be a general class of host shutoff escapees. These results were echoed during lytic infection, with global circRNA levels generally unaffected as lytic infection progressed (Fig. 2C). For the gamma-herpesviruses, KSHV and MHV68, we observed a subpopulation of circRNA which were downregulated by infection. This created a tri-modal circRNA distribution which was not observed for HSV-1. The different circRNA expression modalities for alpha-versus gamma-herpesviruses likely reflects the mechanisms by which specific endoRNases are recruited. HSV-1 vhs targets the 5′ untranslated region (UTR) of mRNAs near the 5′ cap and binding is mediated by the translation factors eIF4H and eIF4AI/II (Feng et al, 2005; Karr and Read, 1999; Page and Read, 2010; Shiflett and Read, 2013). CircRNAs may escape vhs degradation as they do not have a 5′ cap or UTR. This mechanism would not apply to gamma-herpesvirus endoRNases (BGLF5, SOX, muSOX) which bind to a degenerate motif with little preference for relative position within the transcript (Abernathy and Glaunsinger, 2015; Gaglia et al, 2015). Motif scanning suggested high frequency of the consensus "UGAAG" may be partially responsible for host circRNA up- or down-regulation during KSHV lytic reactivation (Table 1). This, however, does not rule out the possibility that SOX targeting may be ameliorated by additional factors. An RNA motif called the SOX-resistant element (SRE) can confer resistance to cleavage in a cis-acting, dominant fashion (Hutin et al, 2013; Muller and Glaunsinger, 2017). CircRNAs tend to shape characteristic stem structures which may potentially act as SREs (Liu et al, 2019). RNA modifications or general accessibility may also confer endoRNase resistance. The host is known to use m6A modifications to

distinguish self and non-self circRNA (Chen et al, 2019). Recently, a m6A reader *YTHDC2* was found to mediate resistance to KSHV SOX (Macveigh-Fierro et al, 2022). RNA Binding Proteins (RBPs) may also play a role in circRNA decay as they can aggregate and cover up to 100% of circRNAs (Okholm et al, 2020), potentially restricting endoRNase substrate recognition. A combination of circRNA sequence and structure likely contributes to the subsets of gamma-herpesvirus endoRNase susceptibility.

CircRNA upregulation in disparate virus and cell models hints at a common mode of induction. As a first line of the antiviral response, the host senses and responds to pathogens through pattern recognition receptors to induce interferons and interferon-stimulated genes. In line with this, the most significantly upregulated pathways during lytic infection were immune-related with IFN-β and -γ predicted as upstream regulators (Appendix Fig. S5). We profiled expression after type I and II interferon treatment and identified 67 upregulated circRNAs (Fig. 4), with half of these also upregulated in our infection models. mRNA and circRNA products for several genes including *EPSTI1*, *B2M*, and *ZCCHC2* were induced comparably after interferon stimulation. These interferon-stimulated circRNAs are therefore likely regulated at the level of gene expression. ISCs with disparate expression changes to their colinear gene product (Fig. EV5A), may be the results of secondary effects such as altered back-splicing or decay. Zinc finger antiviral protein (ZAP) is upregulated upon IFN stimulation and recruits de-capping enzyme Dcp1 and deadenylase PARN to degrade mRNAs (Abernathy and Glaunsinger, 2015). circRNAs are devoid of cap structures or accessible poly(A) tails and likely to resistant to ZAP-mediated degradation, which may result in the accumulation of circRNAs compared to colinear mRNAs upon IFN stimulation.

circRELL1 exemplifies the functionally conserved circRNA. circRELL1 expression was induced by recombinant interferons (β and γ) and the pathogen-associated molecular pattern (PAMP), LPS. Of the stimuli tested, circRELL1 expression was largely unaffected by poly I:C and CpG treatment (Fig. EV4). Poly I:C and CpG DNA are recognized by endosomal toll-like receptors (TLR), TLR3 and TLR9, whereas LPS is sensed by TLR4 on the cell surface. The subcellular localization of TLRs and relative amplitude of their downstream signaling may impact circRELL1 upregulation. In addition, poly I:C treatment can induce robust RNase L-dependent circRNA decay (Liu et al, 2019); a phenomenon that may disguise poly I:C dependent circRELL1 upregulation. We also observed upregulation of circRELL1 following beta- (HCMV) and gamma-herpesvirus (EBV, KSHV) infection. We previously showed that circRELL1 has an anti-lytic cycle role during infection with a gamma-herpesvirus, KSHV (Tagawa et al, 2018; Tagawa et al, 2023). Here, comparable defects in infectious progeny after circRELL1 perturbation were observed for the alpha-herpesvirus, HSV-1 (Fig. 5). This functional conservation signifies the importance of commonly regulated circRNAs identified in this study. The mechanism by which this circRNA represses lytic infection is not fully understood. circRELL1 was found to interact with *TTI1* mRNA, a component of mTOR complex, which is targeted by EBV, KSHV, HCMV, and HSV-1 to regulate viral replication (Le Sage et al, 2016). Effects of infection-induced circRNAs on mTOR signaling pathway and its consequences thus warrant further study.

# Methods

## Reagents and tools

See Table 2.

## Methods and protocols

### Biosafety

All experiments using viruses were performed at the National Institutes of Health (NIH) in Bethesda, Maryland, in Biosafety Level (BSL) 2+ facilities. The BSL-2+ laboratory is equipped with facilities ensuring enhanced safety, such as biological safety cabinets, personal protective equipment, and protocols for handling infectious materials. Research protocols were reviewed and received approval from the Division of Occupational Health and Safety, Office of Research Services at NIH, in accordance with the NIH Guidelines for Research and the Department of Health and Human Services (USA).

### Cells and viruses

Vero (ATCC #CCL-81) were maintained in Dulbecco's modified eagle medium (DMEM, Gibco #11965-092) supplemented with 5% fetal bovine serum (FBS, Gibco #16000044), 1 mM sodium pyruvate (Gibco # 11360070), 2 mM L-glutamine (Gibco # A2916801), 100 U/mL penicillin-streptomycin (Gibco # 15070063). MRC-5 (ATCC #CCL-171) were maintained in DMEM (Gibco) supplemented with 10% FBS, 1 mM sodium pyruvate, 2 mM L-glutamine, 100 U/mL penicillin-streptomycin. NIH 3T3 (ATCC #CRL-1658) and NIH 3T12 (ATCC #CCL-164) were maintained in DMEM (Corning #10-017-CV) supplemented with 8% FBS, 2 mM L-glutamine, 100 U/mL penicillin-streptomycin. A20 HE-RIT B cells harbor a recombinant MHV68 expressing a hygromycin-eGFP cassette (Forrest and Speck, 2008) and doxycycline-inducible RTA (Santana et al, 2017). A20 HE-RIT were maintained in RPMI supplemented with 10% FBS, 100 U/mL penicillin-streptomycin, 2 mM L-glutamine, 50 µM beta-mercaptoethanol, 300 µg/ml hygromycin B, 300 µg/mL gentamicin, and 2 µg/mL puromycin. iSLK-BAC16 (Brulois et al, 2012) were maintained in DMEM (Gibco #11965-092) supplemented with 10% Tet system-approved FBS (Takara #631368), 50 µg/mL hygromycin B, 0.1 mg/mL gentamicin, 1 µg/mL puromycin, 100 U/mL penicillin-streptomycin. 293T (ATCC #CRL-3216) cells were maintained in DMEM supplemented with 10% FBS and 100 U/mL penicillin-streptomycin. Lymphoma cell lines, EBV-positive and negative Akata (designated with (+) or (-)) (Komano et al, 1998), Daudi (ATCC #CCL-213), and BJAB (DSMZ #ACC757) were maintained in Roswell Park Memorial Institute (RPMI) 1640 (Gibco #11875093) supplemented with 10% FBS and 100U/ml penicillin-streptomycin. HDLEC (PromoCell #C-12216) were maintained in EBM-2 basal medium (Lonza #CC-3156) supplemented with EGM-2 SingleQuots supplements (Lonza #CC-4176). Cell culture models were periodically subjected to mycoplasma testing by PCR or enzymatic reporter assays and found negative.

### Virus stock preparation and titration

**HSV-1**. Vero cells were infected with KOS (Smith, 1964) or strain 17 at a low multiplicity of infection (MOI; ~0.01 plaque forming units (PFU)/cell) and harvested when cells were sloughing from the sides of the vessel. Supernatant and cell fraction were collected and centrifuged at $4000 \times g$ 4 °C for 10 min. The subsequent supernatant fraction was reserved. The pellet fraction was freeze ($-80$ °C 20 min)/thawed (37 °C 5 min) for three cycles, sonicated for 1 min, and centrifuged at $2000 \times g$ 4 °C for 10 min. The final virus stock was composed of the cell-associated virus and reserved supernatant virus fractions. Viral stocks were titered by plaque assay on Vero cells.

**KSHV**. iSLK-BAC16 cells were induced with 1 µg/mL doxycycline and 1 mM sodium butyrate for 3 days. Cell debris was removed from the supernatant fraction by centrifuging at $2000 \times g$ 4 °C for 10 min and filtering with a 0.45 polyethersulfone membrane. Virus was concentrated after a $16,000 \times g$ 4 °C 24 h spin and resuspended in a low volume of DMEM media (~1000-fold concentration). To assess viral infectivity, LECs were infected with serial dilutions of BAC16 stock and assessed using CytoFlex S (Beckman Coulter) for GFP+ cells at 3 days post infection. BAC16 contains a constitutively expressed GFP gene within the viral genome. Based on these assays, BAC16 stock was used at a 1:60 dilution, resulting in 70% infection for LEC (MOI 1).

**MHV68**. NIH 3T12-based cell lines were infected at a low MOI with MHV68 until 50% cytopathic effect was observed. Infected cells and conditioned media were dounce homogenized and clarified at $600 \times g$ 4 °C for 10 minutes. Clarified supernatant was further centrifuged at $3000 \times g$ 4 °C for 15 min and then $10,000 \times g$ 4 °C for 2 h to concentrate 40-fold in DMEM. H2B-YFP was prepared and titered using plaque assays in NIH 3T12 cells.

### De novo infection

**HSV-1**. Confluent MRC-5 cells were infected with 0.1 or 10 PFU per cell. Virus was adsorbed in PBS for 1 h at room temperature. Viral inoculum was removed, and cells were washed quickly with PBS before adding on DMEM media supplemented with 2% FBS. Zero-hour time point was considered after adsorption of infected monolayers when cells were place at 37 °C.

**KSHV**. Subconfluent LEC were infected with BAC16 at an approximate MOI of 1 (70% cells infected), as assessed by GFP+ cells at 3 dpi. Virus was adsorbed in a low volume of media containing 8 µg/mL polybrene for 8 h at 37 °C, after which viral inoculum was removed and replaced with fresh media. Zero-hour time point was when virus was added and cells were first placed at 37 °C.

**MHV68**. Subconfluent NIH3T3 fibroblasts were infected with 5 PFU per cell. Virus was adsorbed in a low volume of DMEM media supplemented with 8% FBS for 1 h at 37 °C, prior to overlay with fresh media. Zero-hour time point was when virus was first added and cells were placed at 37 °C.

### HSV-1 mouse infections

Female 8-week-old BALB/cAnNTac (Taconic Biosciences) mice were infected with $10^5$ PFU HSV-1 (strain 17) via the ocular route. The appropriate number of animals for a given experiment were purchased and maintained in microisolator cages at a density of 5 mice per cage. Sample size calculation was not performed. Mice were randomized prior to infection with HSV-1 and were subsequently randomly sampled for tissue harvest. Latently infected trigeminal ganglia were harvested approximately four weeks after primary infection and immediately processed. For explant-induced reactivation, latently infected trigeminal ganglia

**Table 2.  Reagents and tools.**

| Reagent/Resource | Reference or source | Identifier or catalog number |
|---|---|---|
| **Experimental models** | | |
| Vero cells (*C. aethiops*) | ATCC | CCL-81 |
| MRC-5 cells (*H. sapiens*) | ATCC | CCL-171 |
| A20 HE-RIT (*M. musculus*) | Santana et al, 2017 *Pathogens* | N/A |
| iSLK-BAC16 cells (*H. sapiens*) | Brulois et al, 2012 *Journal of Virology* | N/A |
| 293T cells (*H. sapiens*) | ATCC | CRL-3216 |
| Akata, EBV-negative (-) cells (*H. sapiens*) | Komano et al, 1998 *Journal of Virology* | N/A |
| Daudi cells (*H. sapiens*) | ATCC | CCL-213 |
| BJAB cells (*H. sapiens*) | DSMZ | ACC757 |
| HDLEC cells (*H. sapiens*) | PromoCell | C-12216 |
| BALB/cAnNTac (*M. musculus*) | Taconic Biosciences | N/A |
| **Recombinant DNA** | | |
| pCMV-Thy1.1-F2A-dsGFP | Rodriguez et al, 2019 *Journal of Virology* | N/A |
| pCMV-Thy1.1-F2A-vhs | Rodriguez et al, 2019 *Journal of Virology* | N/A |
| pCMV-Thy1.1-F2A-BGLF5 | Rodriguez et al, 2019 *Journal of Virology* | N/A |
| pCMV-Thy1.1-F2A-SOX | Rodriguez et al, 2019 *Journal of Virology* | N/A |
| pCMV-Thy1.1-F2A-muSOX | Rodriguez et al, 2019 *Journal of Virology* | N/A |
| **Antibodies** | | |
| Mouse monoclonal anti-Thy1.1, Alexa Fluor 647 conjugated | BioLegend, clone OX-7 | 202526 |
| **Oligonucleotides and other sequence-based reagents** | | |
| GAPDH_F | IDT | CAGAACATCATCCCTGCCTCTACT |
| GAPDH_R | IDT | GCCGAGCTTCCCGTTCA |
| UL23 (tk)_F | IDT | ACCCGCTTAACAGCGTCAACA |
| UL23 (tk)_R | IDT | CCAAAGAGGTGCGGGAGTTT |
| US1 (ICP22)_F | IDT | TTTGGGGAGTTTGACTGGAC |
| US1 (ICP22)_R | IDT | CAGACACTTGCGGTCTTCTG |
| UL44 (gC)_F | IDT | GTGACGTTTGCCTGGTTCCTGG |
| UL44 (gC)_R | IDT | GCACGACTCCTGGGCCGTAACG |
| GFP_F | IDT | GAAGCAGCACGACTTCTTCAA |
| GFP_R | IDT | AGTCGATGCCCTTCAGCTC |
| vhs_F | IDT | ATCCAACACAATATCACAGCCCATCAACAG |
| vhs_R | IDT | CGCCAACCTCTATCACACCAACACG |
| BGLF5_F | IDT | GACCTCTTGCATGGCCTCTT |
| BGLF5_R | IDT | GTGCCGCTTCAAGTACCTCT |
| SOX_F | IDT | TGGGCGAGTTTATTGGTAGTGAGG |
| SOX_R | IDT | CTCCACTAGACAGCAGATGTGG |
| muSOX_F | IDT | CCCTGGACACTGCTTTCAAT |
| muSOX_R | IDT | GCCTCACAGGGGTTTTTGTA |
| 18S rRNA_F | IDT | GTAACCCGTTGAACCCCATT |
| 18S rRNA_R | IDT | CCATCCAATCGGTAGTAGCG |
| ISG15_F | IDT | GAGAGGCAGCGAACTCATCT |
| ISG15_R | IDT | CTTCAGCTCTGACACCGACA |

    

**Table 2.** (continued)

| Reagent/Resource | Reference or source | Identifier or catalog number |
|---|---|---|
| U6_F | IDT | GGAATCTAGAACATATACTAAAATTGGAAC |
| U6_R | IDT | GGAACTCGAGTTTGCGTGTCATCCTTGCGC |
| 7SL_F | IDT | CAAAACTCCCGTGCTGATCA |
| 7SL_R | IDT | GGCTGGAGTGCAGTGGCTAT |
| MALAT1_F | IDT | GTCATAACCAGCCTGGCAGT |
| MALAT1_R | IDT | GCTTATTCCCCAATGGAGGT |
| SHFL_F | IDT | GGCTCTGATGAGGAAATTCGGC |
| SHFL_R | IDT | TCCTGGGCATCCAGATCGTTAC |
| matGAPDH_F | IDT | AGCCTCAAGATCATCAGCAATG |
| matGAPDH_R | IDT | ATGGACTGTGGTCATGAGTCCTT |
| RPS13_F | IDT | TCGGCTTTACCCTATCGACGCAG |
| RPS13_R | IDT | ACGTACTTGTGCAACACCATGTGA |
| ECE1_F | IDT | CGGAGCACGCGAGCTATGA |
| ECE1_R | IDT | CCGCCAGAAGTACCACCAAC |
| EPSTI1_F | IDT | GACAGAAGTGCCTGTCAAAGTG |
| EPSTI1_R | IDT | GCCGTTTCAGTTCCAGTAATTC |
| IPO7_F | IDT | TCAATTTTGGAGGCCCAGCA |
| IPO7_R | IDT | AGGTTGCATGATCAGGTCCC |
| MAN1A2_F | IDT | CACATGATGATGTACAGCAGAGC |
| MAN1A2_R | IDT | ATCGAACAGCAGGATTACCTGA |
| RELL1_F | IDT | GCAGTGGCACAGAGTAGCAG |
| RELL1_R | IDT | CAGTGCAGCCTTACCAGTTG |
| TNPO3_F | IDT | ACATTGCAGCTCGTGTACCA |
| TNPO3_R | IDT | AGCATGACTCCACATCCTGC |
| circECE1_F | IDT | ACCTCTGGGAACACAACCAA |
| circECE1_R | IDT | GCCGTTGGGGTATGCGTC |
| circEPSTI1_F | IDT | AAGCTGAAGAAGCTGAACTC |
| circEPSTI1_R | IDT | GTGTATGCACTTGTGTATTGC |
| circIPO7_F | IDT | GGGCCAGATGAAGAAGGTAGT |
| circIPO7_R | IDT | GAGCCTGCATTACAGGTCTGAT |
| circMAN1A2_F | IDT | TAGGACATATGGGTGGGGAC |
| circMAN1A2_R | IDT | TGCTTCTTCCAAGGCCTTCT |
| circRELL1_F | IDT | ATGTCTGTTAGTGGGGCTGA |
| circRELL1_R | IDT | TATCTGCTACCATCGCCTTT |
| circTNPO3_F | IDT | GCTAATCGGCGCACAGAAAT |
| circTNPO3_R | IDT | GGTCTGAGATCTCCCATGCA |
| CpG DNA | IDT, RNase Free HPLC Purification, * Phosphorothioate Bond | T*C*G*T*C*G*T*T*T*T*G*T*C*G*T*T*T*T*G*T*C*G*T*T |
| siCircRELL1 | Horizon Life Sciences | AGUAGCAGCGAAUGCUGAUGUUU |
| **Chemicals, enzymes and other reagents** | | |
| JQ1 | Cayman Chemical | CAS: 1268524-70-4 |
| Human IFN-β | Peprotech | 300-02BC |
| Human IFN-γ | Peprotech | 300-02 |
| Transporter 5 | Polysciences | 26008 |
| CD90.1 MicroBeads | Miltenyi | 130-121-273 |

**Table 2.** (continued)

| Reagent/Resource | Reference or source | Identifier or catalog number |
| --- | --- | --- |
| Direct-zol RNA miniprep kit | Zymo | R2053 |
| ReverTra Ace qPCR RT master mix | Toyobo | FSQ-101 |
| Thunderbird Next SYBR qPCR mix | Toyobo | QPX-201 |
| TruSeq Stranded Total RNA Ribo-Zero Gold | Illumina | RS-122-2303 |
| Stranded Total RNA Prep with Ribo-Zero Plus | Illumina | 20040525 |
| ERCC spike-in controls | ThermoFisher | 4456740 |
| **Software** | | |
| Cutadapt | M. Martin 2011 *EMBnet.journal* | N/A |
| STAR | Dobin et al, 2012 *Bioinformatics* | N/A |
| CIRCExplorer3 (CLEAR) | Ma et al, 2019 *Genomics Proteomics Bioinformatics* | N/A |
| RankProd R package | Del Carratore et al, 2017 *Bioinformatics* | N/A |
| FIMO version 5.5.4 | Grant et al, 2011 *Bioinformatics* | N/A |
| FlowJo v.10.9.0 | BD | N/A |
| miEAA v.2.0 | Kern et al, 2020 *Nucleic Acids Research* | N/A |
| Cytoscape 3.10 | Shannon et al, 2003 *Genome Research* | N/A |
| DIANA-TarBase v8.0 | Karagkouni et al, 2018 *Nucleic Acids Research* | N/A |
| WebGestalt | Liao et al, 2019 *Nucleic Acids Research* | N/A |
| Ingenuity Pathway Analysis v.1.22.01 | Qiagen | N/A |
| **Other** | | |
| Illumina NextSeq 550 | Illumina | N/A |
| Illumina NovaSeq SP | Illumina | N/A |
| CytoFlex S | Beckman Coulter | N/A |
| StepOnePlus real-time PCR system | ThermoFisher | N/A |

were explanted into culture (DMEM/1% FBS) for 12 h at 37 °C/5% CO$_2$ in the presence of vehicle (DMSO) or 2 μM JQ1+ to enhance reactivation (Cayman Chemical CAS: 1268524-70-4). Isolated sensory ganglia were pooled and randomized into groups of 6 prior to RNA isolation. Pooled ganglia were homogenized in 1 ml TriPure isolation reagent (Roche) using lysing matrix D on a FastPrep24 instrument (3 cycles of 40 seconds at 6 m/s). 0.2 ml chloroform was added for phase separation using phase lock gel heavy tubes and RNA isolation from the aqueous phase was obtained by using ISOLATE II RNA Mini Kit (Bioline). All animal care and handling were done in accordance with the U.S. National Institutes of Health Animal Care and Use Guidelines and as approved by the National Institute of Allergy and Infectious Diseases Animal Care and Use Committee (Protocol LVD40E, T.M.K.). Samples were identified during RNA-Seq analysis (no blinding).

### Lytic reactivation
**KSHV**. Subconfluent monolayers of iSLK-BAC16 were induced with 1 μg/mL doxycycline, 1 mM sodium butyrate in DMEM media supplemented with 2% Tet system-approved FBS. Zero-hour time point was when induction media was added and cells were first placed at 37 °C.

**MHV68**. One day prior to induction, A20 HE-RIT cells were subcultured at a 1:3 dilution in media lacking antibiotics. Cells were seeded subconfluently and induced for 24 h with RPMI media containing 5 μg/ml doxycycline and 20 ng/ml 12-O-tetradecanoyl-phorbol-13-acetate (TPA).

### rRNA-depleted total RNA-Seq
Total RNA was isolated from cells using the Direct-zol RNA MiniPrep kit (Zymo Research R2053), following manufacturer's instructions. ERCC spike-in controls (ThermoFisher 4456740) were added to 500–1000 ng of total RNA. RNA was sent to the NCI CCR-Illumina Sequencing facility for library preparation and sequencing. RNA was rRNA depleted and directional cDNA libraries were generated using either Stranded Total RNA Prep with Ribo-Zero Plus (Illumina # 20040525) or TruSeq Stranded Total RNA Ribo-Zero Gold (Illumina #RS-122-2303). 2–4 biological replicates were sequenced for all samples. Sequencing was

performed at the National Cancer Institute Center for Cancer Research Frederick Sequencing Facility using the Illumina NextSeq 550 or Illumina NovaSeq SP platform to generate 150 bp paired-end reads.

### RNA extraction and RT-qPCR

Total RNA was extracted with Direct-zol RNA miniprep kit with on-column DNase I digestion (Zymo Research #R2053). 0.5 to 1 µg of total RNA was used for reverse-transcription with ReverTra Ace qPCR RT master mix (Toyobo #FSQ-101) and quantitative PCR (qPCR) was performed with Thunderbird Next SYBR qPCR mix (Toyobo #QPX-201) and StepOnePlus real-time PCR system (ThermoFisher) following manufacturer's instructions.

### Measuring viral genomes

The cell fraction was isolated from infection models. Cell pellets were washed with 1x PBS and lysed using 0.5% SDS, 400 µg/mL proteinase K, 100 mM NaCl. Samples were incubated at 37 °C for 12–18 h and heat inactivated for 30 minutes at 65 °C. DNA samples were serial diluted 1:1000 and measured using qPCR with primers specific to HSV-1 UL23 and human *GAPDH*. Standard curves were generated using purified genomic stocks (HSV-1 bacterial artificial chromosome and human genome Promega #G1471). Absolute copy number of genomic stocks was determined using droplet digital PCR (Biorad QX600). Values were plotted as follows:

$$\text{viral genomes/cell} = \frac{\text{viral gene copy number}}{\text{host gene copy number}/2}.$$

### Viral nuclease ectopic expression

$3 \times 10^6$ 293T cells were seeded to 10 cm petri dishes and incubated overnight. Cells were transfected with 8 µg of plasmid DNAs (pCMV-Thy1.1-F2A-dsGFP/muSOX/SOX/vhs/BGLF5) using 48 µl Transporter 5 (Polysciences #26008) and 1 ml Opti-MemI (Gibco #31985070). After 24 h, cells were resuspended in 5 ml staining buffer [1x PBS Gibco # 10010023 supplemented with 2 mM ethylenediaminetetraacetic acid (Sigma) and 0.5% FBS (Gibco)] and mouse Thy1.1-expressing cells were magnetically enriched with CD90.1 MicroBeads (Miltenyi #130-121-273). Cells were stained with Alexa Fluor 647 anti-mouse Thy-1.1 Antibody (BioLegend; clone OX-7) and enrichment was confirmed with CytoFlex S (Beckman Coulter). 70–80% of cells were positive for Thy1.1 after sorting and lysed with TRI reagent (Zymo Research #R2050-1) for RT-qPCR.

### Immune stimulation of cell culture models

Confluent monolayers of MRC-5 and LEC were treated with a variety of immunostimulants. 10 µg/mL lipopolysaccharide (Millipore Sigma L3024), 4 µg/mL CpG DNA (IDT), 10 µg/mL poly I:C (Millipore Sigma #P1530), or 25 ng/mL recombinant human IFN-β and γ (Peprotech #300-02BC and #300-02) was added to culture media. After the indicated treatment time (8, 24, 48, or 72 h), RNA was isolated from the cell fraction. For Akata-, Daudi, or BJAB, 10 µg/mL lipopolysaccharide, 10 µg/mL poly I:C, 10 ng/mL IFN-β, or 500 ng/mL IFN-γ were added to culture media. After 24 h RNA was isolated from the cell fraction.

### Interferon stimulation

Confluent monolayers of MRC-5 and LEC were treated with 25 ng/mL recombinant human IFN-β and γ (Peprotech #300-02BC)

and #300-02) in the culture media. For Akata(-) 10 ng/mL IFN-β was added to culture media. After 48 h RNA was isolated from the cell fraction. Human peripheral blood mononuclear cells (PBMCs) were prepared from buffy coats. Buffy coats from two different donors were purified with Ficoll-Paque PLUS (GE Healthcare #17144003). PBMCs were treated with red blood cell lysis buffer (BioLegend #420301) and washed three times with phosphate-buffered saline (Gibco #10010023). 10 or 30 ng/mL recombinant human IFN-β (Peprotech #300-02BC) was added to the culture media. After 24 h RNA was isolated from the cell fraction.

### Immunostaining and flow cytometry

Cells were washed with PBS, detached with Accutase (BioLegend), and mixed with antibodies in staining buffer (PBS, 0.5% FBS, 2 mM EDTA) for 20 min at room temperature. Alexa Fluor 647 anti-mouse Thy-1.1 Antibody (BioLegend; clone OX-7) was used. Fluorescence was measured by CytoFlex S (Beckman Coulter) flow cytometer and analyzed with FlowJo 10.9.0 (BD). Doublets were removed and Allophycocyanin (APC) channel was used to quantify Thy1.1 levels at cell surface.

### In silico circRNA-miRNA-mRNA network predictions

Four commonly-regulated human circRNAs (hsa_circ_0001730, hsa_circ_0006990, hsa_circ_0006646, hsa_circ_0001769) were examined for miRNA-binding sites using CircInteractome (Dudekula et al, 2016). Experimentally supported miRNA-mRNA interactions were then added to the network based on DIANA-TarBase v8.0 (Karagkouni et al, 2018). Cytoscape 3.10 was used for visualizing the network (Shannon et al, 2003). Overrepresentation analysis of predicted target miRNAs was performed with miEAA 2.0 (Kern et al, 2020). Enriched Gene Ontology terms (http://geneontology.org), p-values, and miRNAs that were predicted to be regulated by circRNAs are shown.

### Overrepresentation analysis (ORA)

ORA was performed using a list of genes colinear to circRNAs detected in a given model (>10 raw BSJ read counts). ORA was performed using WebGestalt (Liao et al, 2019) and plotted using SRplot (https://www.bioinformatics.com.cn/en).

### Ingenuity pathway analysis (IPA)

IPA (Qiagen) was performed on de novo lytic infection RNA-Seq datasets for HSV-1 (MRC-5 infected at MOI (multiplicity of infection) of 10 for 12 h, 4 biological replicates), HCMV (MRC-5 infected at MOI of 3 for 72 h, 2 biological replicates), and KSHV (LEC infected at MOI of 1 for 72 h, 2 biological replicates). $\log_2$ fold change ($\log_2$FC, Infected/Uninfected) was calculated from ERCC normalized data. Only genes with an average $\log_2$FC > 1 were used for IPA. Bubble plots were generated using SRplot (https://www.bioinformatics.com.cn/en).

### circRELL1 functional studies

**Loss of function.** Subconfluent MRC-5 were incubated for 8 h at 37 °C in OptiMEM (ThermoFisher #31985062) with 20 nM siRNA and Lipofectamine RNAiMAX Transfection Reagent (ThermoFisher #13778150). siRNAs were ON-TARGETplus Non-targeting Control Pool (Horizon #D-001810-10-05) and ON-TARGETplus siCircRELL1,

previously published as "siCirc1400-2" (Tagawa et al, 2018). This siRNA target back-splice junctions of circRELL1 (circ_0001400) and does not affect the linear counterpart RNA, *RELL1*. After 8 h transfection media was removed and cells incubated in 1x DMEM supplemented with 2% FBS, 1 mM sodium pyruvate, 2 mM L-glutamine for an additional 40 h at 37 °C. If indicated, cells were infected with HSV-1 strain KOS at an MOI of 0.1 or 10 PFU/cell and collected at 12 h post infection.

**Gain of function.** Lentiviral vectors expressing circGFP or circRELL1 with a fluorescent (mCherry) reporter were cloned and pseudotyped with VSV-G were cloned and purified by Vector-Builder (Tagawa et al, 2023). The circRELL1 insert contains exons 4–6 of the colinear transcript (NCBI RefSeq NM_001085400.2). Inserts are flanked by *ZKSCAN1* intronic sequences which facilitate circularization of the internal sequence (Kramer et al, 2015). Subconfluent MRC-5 were incubated with lentiviral vectors in 1x DMEM supplemented with 10% FBS, 1 mM sodium pyruvate, 2 mM L-glutamine, 8 μg/mL polybrene. Cells were incubated with lentivirus at 37 °C for 18 h, before media was replaced. MRC-5 were infected with comparable infectious doses of circGFP and circRELL1 lentivirus, such that 70% of cell were mCherry+ at three days post infection. Forty hours after lentivirus infection, MRC-5 were infected with HSV-1 strain KOS at an MOI of 10 PFU/mL and the cell fraction collected at 12 h post HSV-1 infection.

### Gene quantitation

RNA-Sequencing reads were trimmed using Cutadapt (Martin, 2011) and the following parameters: --pair-filter=any, --nextseq-trim=2, --trim-n, -n 5, --max-n 0.5, -0 5, -q 20, -m 15. Trimmed reads were mapped using STAR (Dobin et al, 2012) with 2-pass mapping to concatenated genome assemblies which contain the host genome (hg38 or mm39) + virus genome (KT899744.1, NC_001806.2, NC_006273.2, NC_009333.1, MH636806.1) + ERCC spike-in controls. Details on mapping assemblies are included below. RNA STAR mapping parameters are as follows: --outSJfilterOverhangMin 15 15 15 15, --outFilterType BySJout, --outFilterMultimapNmax 20, --outFilterScoreMin 1, --outFilterMatchNmin 1, --outFilterMismatchNmax 2, --outFilterMismatchNoverLmax 0.3, --outFilterIntronMotifs None, --alignIntronMin 20, --alignIntronMax 2000000, --alignMatesGapMax 2000000, --alignTranscriptsPerReadNmax 20000, --alignSJoverhangMin 15, --alignSJDBoverhangMin 15, --alignEndsProtrude 10 ConcordantPair, --chimSegmentMin 15, --chimScoreMin 15, --chimScoreJunctionNonGTAG 0 --chimJunctionOverhangMin 18, --chimMultimapNmax 10. RNA STAR GeneCount (per gene read counts) files were used for transcript quantitation. The only RNA-Seq data not generated in-house is from HCMV infection (Data ref: Oberstein and Shenk, 2017) (Table 3). This data did not contain ERCC spike-in controls and thus was normalized as Transcripts per Million (TPM). For all other models,

**Table 3. RNA-Seq data analyzed in this study.**

| Model | | | Metadata | Accession |
|---|---|---|---|---|
| HSV-1 | Lytic infection | MRC-5 cells infected with HSV-1 strain KOS (MOI 10 PFU/cell) for 12 or 24 h. PMID: 36724259 | $n = 2$–4 Illumina 150 PE | SRR19779319, SRR19779318, SRR19787559 |
| | Latent infection | Female 8-week-old BALB/cAnTAC mice infected with HSV-1 strain 17 ($10^5$ PFU) via the ocular route. Latently infected trigeminal ganglion (TG) were harvested 4 weeks after primary infection. | $n = 4$ Illumina 150 PE | SRR19792335, SRR19792334 |
| | Lytic reactivation | Female 8-week-old BALB/cAnTAC mice infected with HSV-1 strain 17 ($10^5$ PFU) via the ocular route. Trigeminal ganglia were explanted (TG explant) ~4 weeks after primary infection in the presence of vehicle or 2 μM JQ1 for 12 h. | $n = 2$–3 Illumina 150 PE | SRR25824398, SRR25824397, SRR25824394, SRR25824396 |
| HCMV-Lytic infection | | MRC-5 cells infected with HCMV strain TB40/E (MOI 3 PFU/cell) for 24 or 72 h. PMID: 28874566 | $n = 2$ Illumina 100 PE | SRR5629593, SRR5629594, SRR5629591, SRR5629592, SRR5629589, SRR5629590, SRR5629587, SRR5629588, SRR5629577, SRR5629578, SRR5629575, SRR5629576, SRR5629573, SRR5629574, SRR5629571, SRR5629572 |
| KSHV | Lytic infection | LEC infected with KSHV strain BAC16 (MOI 1 PFU/cell) for 24 or 72 h. PMID: 36724259 | $n = 2$ Illumina 150 PE | SRR20020769, SRR25816558, SRR20020770 |
| | Lytic reactivation | iSLK-BAC16 induced with 1 μg/mL Doxycycline 1 mM Sodium Butyrate for 24 or 72 h. PMID: 36724259 | $n = 4$ Illumina 150 PE | SRR25816557, SRR20020761, SRR25816556, SRR20020757, SRR20020758 |
| MHV68 | Lytic infection | NIH 3T3 cells infected with MHV68 strain H2B-YFP (MOI 5 PFU/cell) for 6 or 18 h. | $n = 2$ Illumina 150 PE | SRR19792326, SRR25823339, SRR19792325 |
| | Lytic reactivation | A20 HE-RIT cells induced with 5 μg/ml Dox and 20 ng/ml TPA for 6 or 24 h. | $n = 2$ Illumina 150 PE | SRR19792324, SRR25823338, SRR19792321 |
| Interferon treatment | | MRC-5, LEC, or Akata- cells treated with IFN-β or -γ or- for 48 h. | $n = 3$ Illumina 150 PE | SRR25905055, SRR25905049, SRR25905048, SRR25905050, SRR25905054, SRR25905051, SRR25905053, SRR25905052 |
| EBV-Latent infection | | Akata+ (EBV positive) or Akata- (EBV negative) cells. 074301 Arraystar Human CircRNA microarray V2. PMID: 36724259 | $n = 3$ Microarray | GSE206824 |

Overview of bulk RNA-Seq data analyzed in this study. If data from any of the sample sets was previously published we have included PubMed reference numbers.

 

ERCC reads were used to generate standard curves similar to (Schertzer et al, 2020), using their known relative concentrations. All biological replicates had ERCC-derived standard curves with $R^2 > 0.9$. ERCC normalized gene counts were calculated as follows:

$$\text{Log}_2\text{RPKM}: \text{Log}_2\left(\frac{\text{Raw gene counts}}{\text{gene size in } kb \times \text{Million total reads}}\right)$$

*Raw gene counts includes forward spliced reads and excludes reads containing BSJ.*

$$\text{ERCC norm.gene counts}: \left(2^{\left(\frac{\text{Log}_2(\text{RPKM}) - \text{ERCC derived intercept } (b)}{\text{ERCC derived slope } (y)}\right)}\right)/10{,}000$$

### CircRNA quantitation

RNA-Seq data was trimmed (Cutadapt) and aligned (STAR, 2-pass) as described in the gene quantitation section above. Note that we required a minimum of 18 nucleotides flanking any chimeric BSJ calls to ensure high-confidence in circRNA quantitation. Back-splice junctions (BSJ) were quantified using CIRCExplorer3 (CLEAR) pipeline (Ma et al, 2019) and normalized as TPM (HCMV data) or relative to ERCC spike in controls (all other data). BSJ variants are reported relative to their colinear gene products and circbase annotations (http://www.circbase.org/) (Glažar et al, 2014). Gene length for circRNA was treated as 0.15 kb as that is the total read length and full circRNA size is unknown. ERCC normalized circRNA counts were calculated as above.

### Differentially expressed circRNA (DEC) calling

**Bulk RNA-Seq data.** Up- or down-regulated circRNAs had a raw BSJ count across the sample set >10, $\log_2\text{FC} > 0.5$, and $p$-value < 0.05. With the exception of interferon-stimulated RNA-Seq data (Fig. 4), significance was calculated by rank product paired-analysis with RankProd R package (Del Carratore et al, 2017). For interferon-stimulated RNA-Seq data EdgeR was used to calculate statistical significance.

**EBV microarray data.** Data was previously published in (Data ref: Tagawa et al, 2023), comparing Akata(+) and Akata(-) cells assessed by microarray (074301 Arraystar Human CircRNA microarray V2). Upregulated circRNAs had a $\log_2\text{FC} > 0.5$ and $p$-value < 0.05. Significance was calculated by rank product paired-analysis with RankProd R package (Del Carratore et al, 2017).

### Analysis of SOX substrate-targeting motif

CircRNA subsets were binned using the average $\log_2$ fold change (iSLK-BAC16 3 days post induction/uninduced) from four biological replicates. CircRNA back splice junctions were cross-referenced to circAtlas (Wu et al, 2020) to determine expected full-length sequences. Motif scanning was performed using FIMO (Grant et al, 2011) version 5.5.4 looking for exact matches to "UGAAG" with the following run parameters: --verbosity 1 --bgfile --nrdb-- --thresh 0.01 --no-pgc.

### Transcript isoform analysis

RNA-Sequencing reads were trimmed using Cutadapt (Martin, 2011) and the following parameters: --pair-filter=any, --nextseq-trim=2, --trim-n, -n 5, --max-n 0.5, -0 5, -q 20, -m 15. Trimmed reads were mapped and quantified using Salmon quant Version 1.10.1 (Patro et al, 2017) using a transcript fasta reference and the following parameters: --kmerLen 31, --incompatPrior 0.0, --biasSpeedSamp 5, --fldMax 1000, --fldMean 250, --fldSD 25, --forgettingFactor 0.65, --maxReadOcc 100, --numBiasSamples 2000000, --numAuxModelSamples 5000000, --numPreAuxModelSamples 5000, --numGibbsSamples 0, --numBootstraps 0, --thinningFactor 16, --sigDigits 3, --vbPrior 1e-05. The reference transcriptome fasta was created using gffread v0.12.7 (Pertea and Pertea, 2020) with the following parameters: -w, -g (reference genome: GRCh38 version 36, Ensembl 102). Salmon transcript and gene quantitation files were normalized to ERCC spike-in reads as detailed in the main Methods section.

### Genome assemblies

HSV-1, strain KOS: KT899744.1, with the corresponding coding sequence (CDS) annotation used for transcript quantification.

HSV-1, strain 17: NC_001806.2, with the corresponding CDS annotation used for transcript quantification.

HCMV: NC_006273.2, with the corresponding CDS annotation used for transcript quantification.

KSHV: NC_009333.1, with the corresponding CDS annotation used for transcript quantification.

MHV68: MH636806.1 (O'Grady et al, 2016) modified to remove the beta-lactamase gene (Δ103,908-105,091), with the corresponding CDS annotation used for transcript quantification.

Human: hg38, gencode.v36.

Mouse: mm39, gencode.vM29.

ERCC spike-in: available from ThermoFisher (#4456740).

## Data availability

Additional information about data analyzed in this study is present in Table 3. HSV-1 infection RNA-Seq data: Sequence Read Archive (SRA) database SRR19779319, SRR19779318, SRR19787559 https://www.ncbi.nlm.nih.gov/sra?term=SRP383035. https://www.ncbi.nlm.nih.gov/sra?term=SRP383161. HSV-1 murine infection RNA-Seq data: SRA database SRR19792335, SRR19792334, SRR25824398, SRR25824397, SRR25824394, SRR25824396 https://www.ncbi.nlm.nih.gov/sra?term=SRP383241. KSHV infection RNA-Seq data: SRA database SRR20020769, SRR20020770, SRR20020761, SRR20020757, SRR20020758, SRR25816558, SRR25816557, SRR25816556 https://www.ncbi.nlm.nih.gov/sra?term=SRP385335. MHV68 infection RNA-Seq data: SRA database SRR19792326, SRR19792325, SRR19792324, SRR19792321, SRR25823338, SRR25823339 https://www.ncbi.nlm.nih.gov/sra?term=SRP383239. Interferon stimulation RNA-Seq data: SRA database SRR25905055, SRR25905049, SRR25905048, SRR25905050, SRR25905054, SRR25905051, SRR25905053, SRR25905052. https://www.ncbi.nlm.nih.gov/sra?term=SRP458394.

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

## Acknowledgements

This work was supported by the Intramural Research Program of the National Cancer Institute (JMZ; 1ZIABC011176, LTK; 1ZIABC011953) and National Institute of Allergy and Infectious Diseases (TMK; 1ZIAAI000712). TT was funded by a research fellowship from the Japan Society for the Promotion of Science. The funders had no role in study design, data collection and analysis, decision to publish, or preparation of the manuscript. We thank the Center for Cancer Research Sequencing Facility (Frederick, MD) for their assistance in sequencing libraries. The resources of the NIH High-Performance Computing BIOWULF Cluster were utilized for all our computational needs. We thank Mandy Muller (University of Massachusetts Amherst) for plasmids containing viral endonucleases. We thank Neal DeLuca (University of Pittsburgh) for HSV-1 (strain KOS). We thank Bill Sugden (University of Wisconsin Madison) for providing B cell lymphoma cell lines.

## Author contributions

**Sarah E Dremel**: Conceptualization; Formal analysis; Investigation; Visualization; Methodology; Writing—original draft; Writing—review and editing. **Takanobu Tagawa**: Conceptualization; Investigation; Methodology; Writing—original draft; Writing—review and editing. **Vishal N Koparde**: Data curation; Software; Writing—review and editing. **Carmen Hernandez-Perez**: Investigation; Writing—review and editing. **Jesse H Arbuckle**: Investigation; Methodology; Writing—review and editing. **Thomas M Kristie**: Investigation; Methodology; Writing—review and editing. **Laurie T Krug**: Investigation; Methodology; Writing—review and editing. **Joseph M Ziegelbauer**: Supervision; Funding acquisition; Writing—review and editing.

## Disclosure and competing interests statement

The authors declare no competing interests.

# Expanded View Figures

**Figure EV1.  Murine circRNAs upregulated in HSV-1 infection models.**

(**A**) Infographic for murine HSV-1 infection models used in this study. (**B**) Overlap of upregulated murine circRNA detected by bulk RNA-Seq from HSV-1 latency (TG, $n = 4$), explant-induced reactivation (TG explant, $n = 3$), and drug-enhanced reactivation (TG explant + JQ1, $n = 2$). (**C**) Heatmaps for upregulated murine circRNAs, plotted as circRNA counts ($\log_2$FC), Gene counts ($\log_2$FC), or CIRCScore (circFPB/linearFPB). Data Information: Bulk RNA-Seq data was normalized to ERCC spike-in controls. $\log_2$FC is relative to a paired uninfected control. Differential expression *p*-values were calculated using RankProd non-parametric permutation tests. DECs were those with raw back splice junction (BSJ) count across the sample set >10, $\log_2$FC > 0.5, and *p*-value < 0.05. Heatmap values are the average of biological replicates.

▶

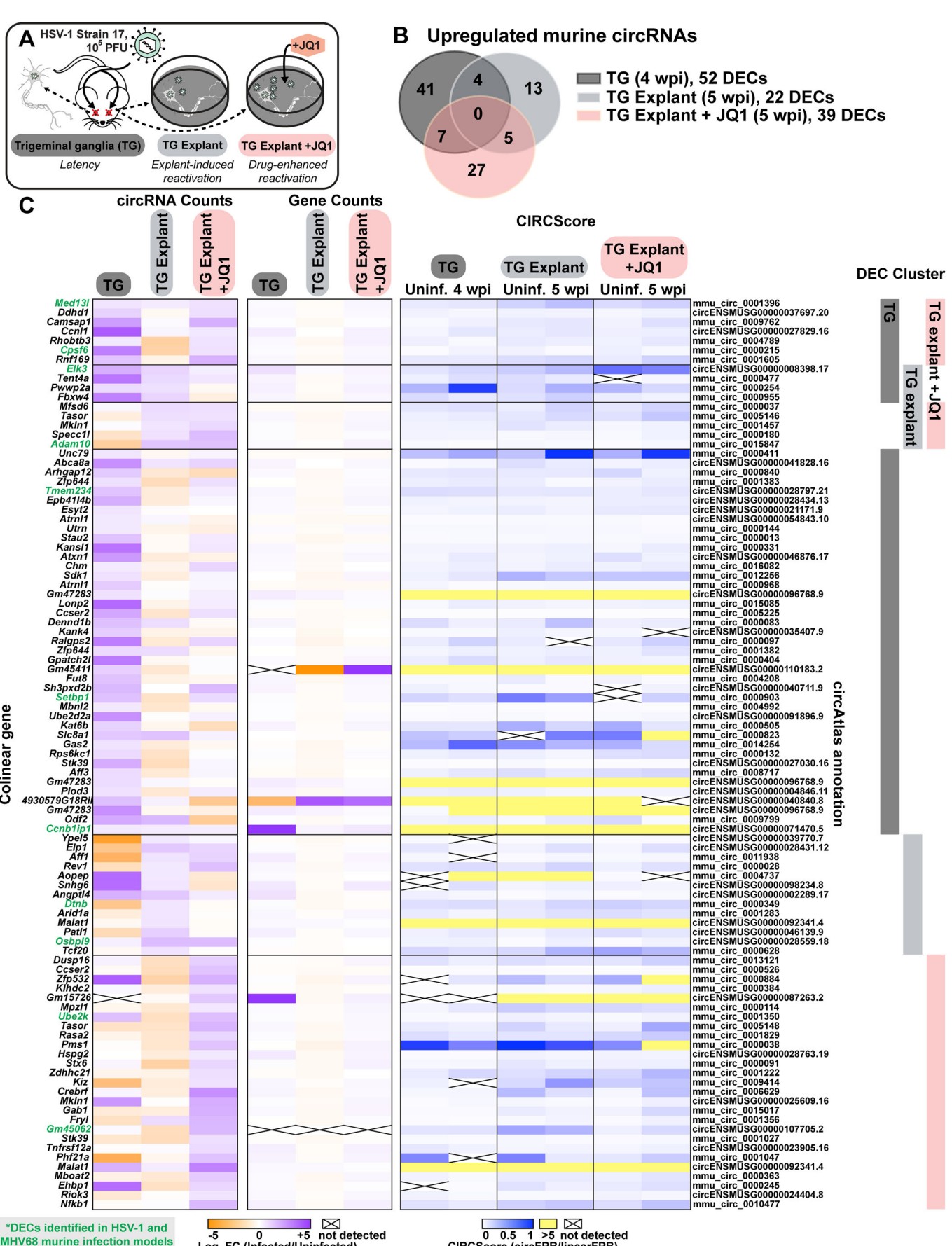

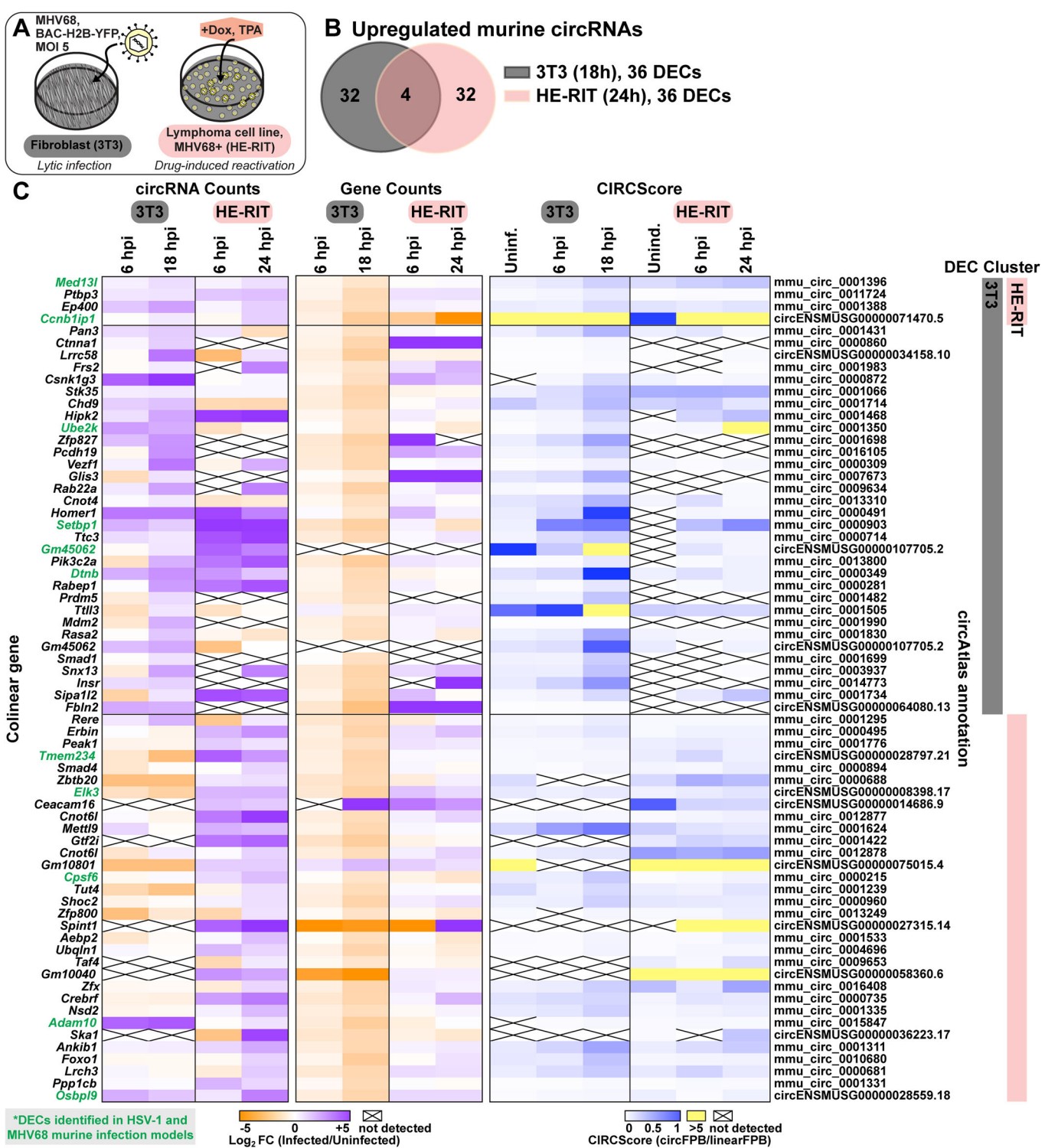

**Figure EV2. Murine circRNAs upregulated in MHV68 infection models.**

(A) Infographic for murine MHV68 infection models used in this study. (B) Overlap of upregulated murine circRNA detected by bulk RNA-Seq from MHV68 de novo infection (NIH 3T3, $n = 2$) and drug-induced reactivation (HE-RIT, $n = 2$). (C) Heatmaps for upregulated murine circRNAs, plotted as circRNA counts (log2FC), Gene counts (log2FC), or CIRCScore (circFPB/linearFPB). Bulk RNA-Seq data was normalized to ERCC spike-in controls. Log2FC is relative a paired uninfected or uninduced control. Differential expression $p$-values were calculated using RankProd non-parametric permutation tests. DECs were those with raw back splice junction (BSJ) count across the sample set >10, log2FC > 0.5, and $p$-value < 0.05. Heatmap values are the average of biological replicates.

**A**

Gene Symbol **GAPDH**   *<-- type your gene of interest here, and graph/table will autopopulate*

| ERCC normalized counts | | | | |
|---|---|---|---|---|
| GAPDH | R1 | R2 | R3 | R4 |
| Uninf. | 23303.5557 | 32095.5317 | 36797.0551 | 26308.1991 |
| 12 hpi | 2349.11904 | 1100.68069 | 2770.04847 | 4144.34788 |
| 24 hpi | | | 1954.29968 | 2043.8145 |
| Log2FC (Infected/Uninfected), pseudozero=0.001 | | | | |
| GAPDH | R1 | R2 | R3 | R4 |
| 12 hpi | -3.3103584 | -4.8659046 | -3.7316072 | -2.6662955 |
| 24 hpi | | | -3.5758265 | -3.9730363 |

**B**

circbase ID **hsa_circ_0001730**   *<-- type your circRNA of interest here, and graph/table will autopopulate*

colinear gene **EPHB4**

| ERCC normalized counts | | | | |
|---|---|---|---|---|
| hsa_circ_000 | R1 | R2 | R3 | R4 |
| Uninf. | NA | 9.95340501 | 6.58956575 | NA |
| 12 hpi | 17.7736569 | 7.97647606 | 11.9430121 | 17.786961 |
| 24 hpi | | | 18.8083533 | 32.1973401 |
| Log2FC (Infected/Uninfected), pseudozero=0.001 | | | | |
| hsa_circ_000 | R1 | R2 | R3 | R4 |
| 12 hpi | 14.1174529 | -0.3194386 | 0.85791144 | 14.1185324 |
| 24 hpi | | | 14.1990859 | 1.69367945 |

◀  **Figure EV3.  Screenshot of interactive resource table for transcriptomic data.**

(A) Representative screenshot of the GenePlotter tab where users can query any gene of interest. Supporting datasets were generated for all bulk RNA-Seq data analyzed in this study. Datasets include raw and normalized gene counts. (B) Representative screenshot of the CircPlotter tab where users can query any circRNA of interest. Supporting datasets were generated for all bulk RNA-Seq data analyzed in this study. Datasets include raw and normalized BSJ counts.

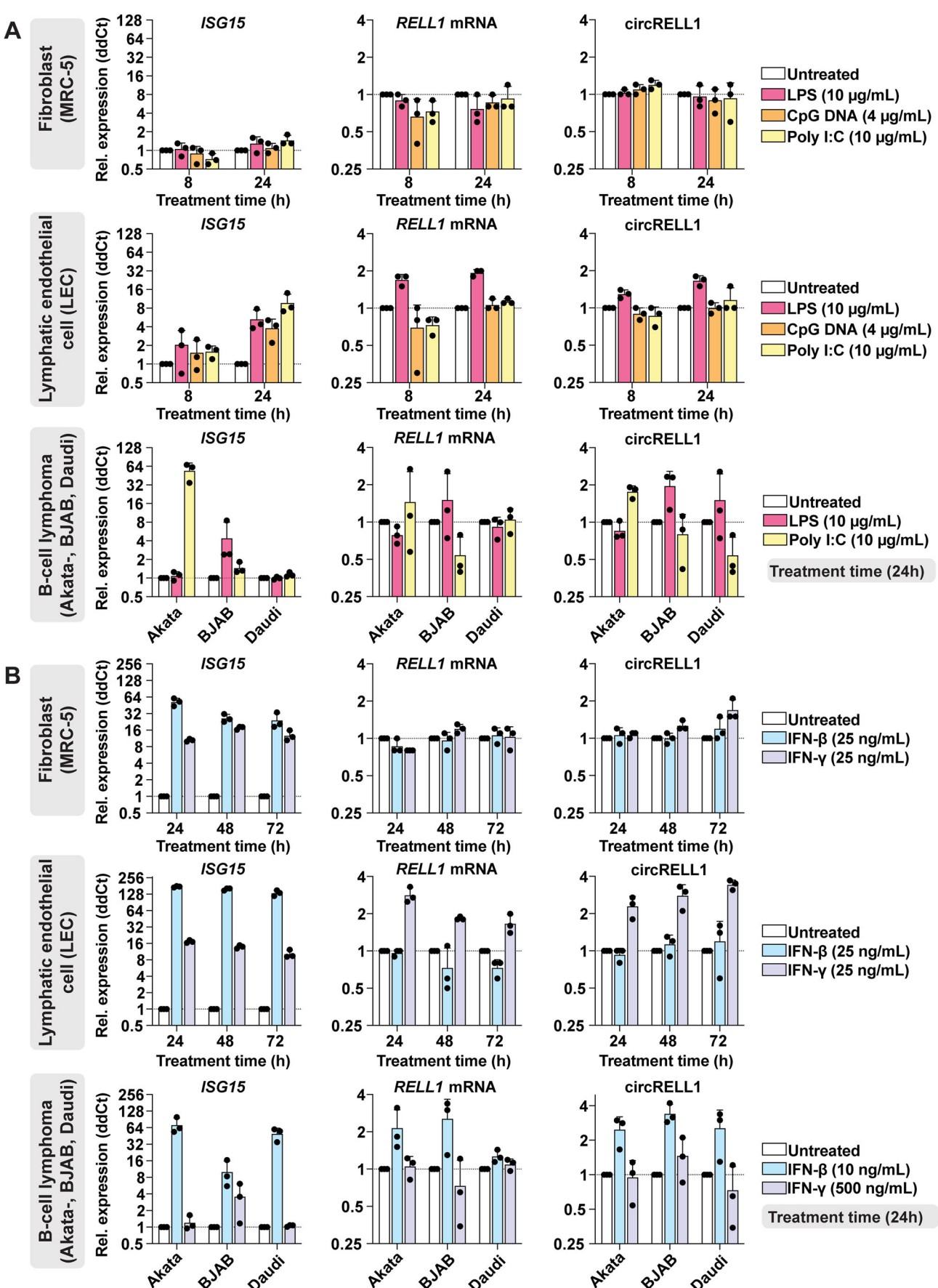

**Figure EV4.** **CircRELL1 expression changes in response to immune stimuli.**

(A) Transcript quantitation for RNA collected from fibroblasts (MRC-5), lymphatic endothelial cells (LEC), or B-cell lymphoma cells (Akata-, BJAB, Daudi) treated with lipopolysaccharide (LPS), poly I:C, or CpG DNA ($n = 3$). (B) Transcript quantitation for RNA collected from MRC-5, LEC, or B-cell lymphoma cells treated with IFN-β and -γ ($n = 3$). Data Information: qPCR data is plotted as relative expression (ddCt) using 18S rRNA as the reference gene, and relative to a paired untreated sample. Data points are biological replicates, column bars are the average, and error bars are standard deviation.

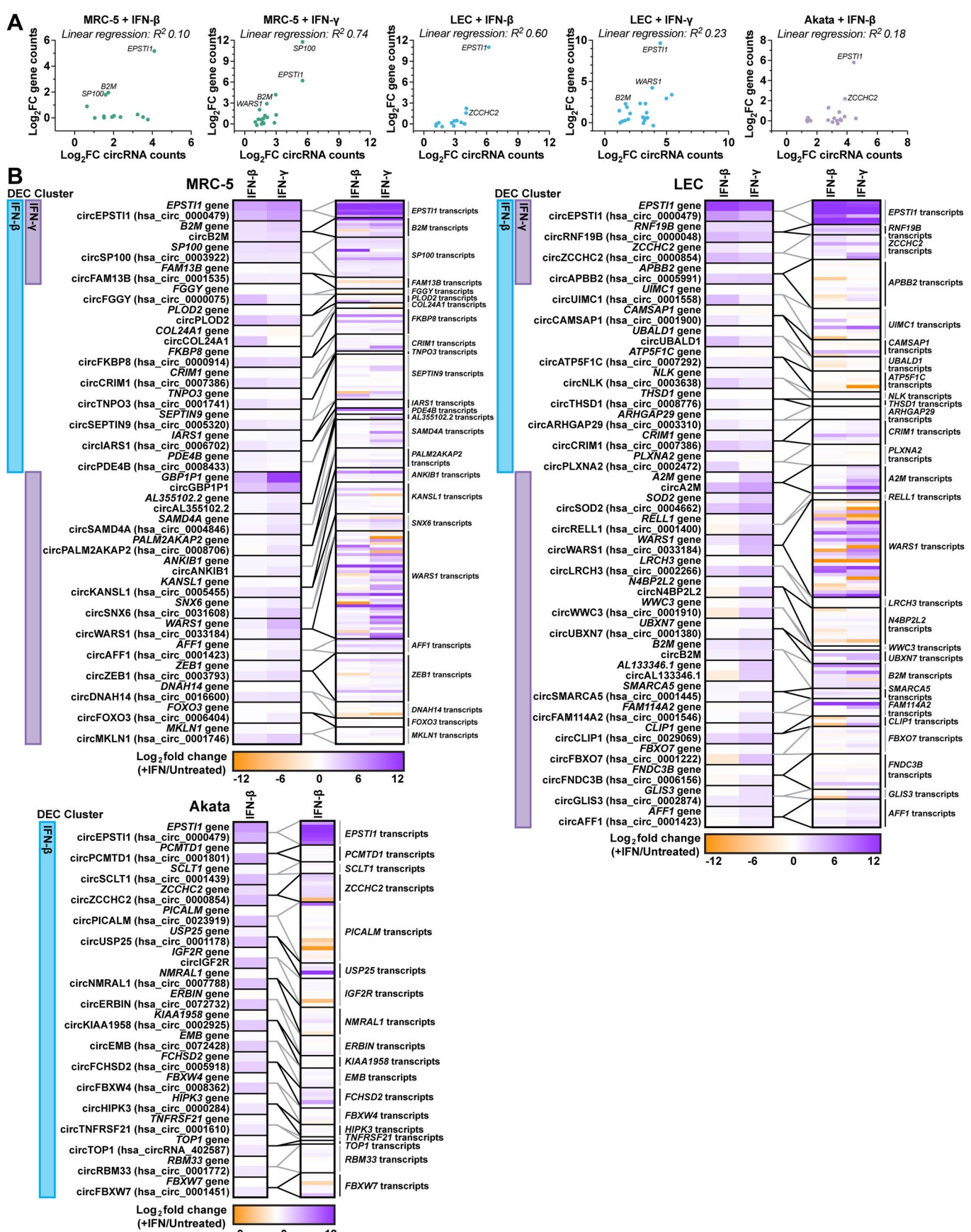

◄ **Figure EV5. Transcript isoform analysis of interferon-stimulated cells.**

(A) Bulk RNA-Seq data from Fig. 4 plotting $\log_2$FC (+IFN/untreated) for DECs relative to their colinear gene reads. Genes with similar levels of upregulation for circRNA and colinear gene species are labeled. (B) Bulk RNA-Seq data from Fig. 4 plotted as heatmaps showing $\log_2$FC (+IFN/untreated) for circRNA species and their colinear gene or transcript isoforms. Gene and transcript quantitation was performed using Salmon and a reference transcriptome fasta. The sample (+IFN-β or +IFN-γ) where a differentially expressed circRNA (DEC) was identified are labeled on the left of each heatmaps as "DEC cluster". Data Information: Bulk RNA-Seq data was normalized to ERCC spike-in control and $\log_2$FC was determined relative to a paired untreated sample. All data plotted is the average of biological replicates ($n = 3$). The relationship between circRNA and colinear gene expression changes was assessed by linear regression analysis, $R^2$ values are given.

