## [58459 Peer Review File · EMBO Reports]

Interferon induced circRNAs escape herpesvirus host shutoff and suppress lytic infection

Sarah Dremel, Takanobu Tagawa, Vishal Koparde, Carmen Hernandez-Perez, Jesse Arbuckle, Thomas Kristie, Laurie T. Krug, and Joseph Ziegelbauer

DOI: [0.15252/embr.202358459](https://doi.org/10.15252/embr.202358459)

Corresponding author: Joseph Ziegelbauer (ziegelbauerjm@mail.nih.gov)

Review Timeline:

Transferred from Review Commons:	9th Nov 23
Editorial Decision:	21st Nov 23
Revision Received:	8th Dec 23
Editorial Decision:	14th Dec 23
Revision Received:	20th Dec 23
Accepted:	21st Dec 23

Editor: Achim Breiling

Transaction Report:

This manuscript was transferred to EMBO Reports following peer review at Review Commons.

Review #1

1. Evidence, reproducibility and clarity:

Evidence, reproducibility and clarity (Required)

The manuscript by Dremel et al reports on IFN-induced circRNAs that are produced in response to herpesviral infections and that are resistant to virus-mediated degradation (e.g. host shut-off). They provide a wealth of interesting data on circRNA production changes in response to infection with alpha, beta, and gamma herpesviruses and identify a number of host circRNAs that are commonly regulated across all subfamilies. Using circRELL1 as an example, they demonstrate a modest impact on productive HSV-1 infections (which follows up on their previous report of circRELL1 impacting KSHV lytic infection). They subsequently propose a new model in which a subset of IFN-stimulated genes produce both mRNAs and circRNAs with the potential for antiviral activity, the latter operating as a 'workaround' for avoiding virus-mediated shutoff.

Major Comments

- Line 146: The authors extended their analysis to HSV-1 latently infected mouse trigeminal ganglia but potentially missed a trick but not including an analysis of HSV-1/VZV infected human trigeminal ganglia for which potentially compatible datasets are available (PMID: 29563516).
- Line 264: The authors note that a direct relationship in gene expression changes was observed between some circRNAs and mRNAs and not others. However, a gene level analysis may be problematic here as many genes encode multiple distinct transcript isoforms that are variably regulated e.g. during infection. A transcript level analysis (e.g. using Kallisto/Salmon) might enable the authors to specifically link circRNAs with individual mRNA isoforms which would be valuable information in the context of interferon-driven gene expression. Alternatively, the lack of a direct relationship in gene expression changes may also link to variability in circRNA decay / halflives. The authors could potentially solve this using a metabolic labelling approach to measure mRNA and circRNA decay rates for a subset of the genes of interest.
- Line 273: The weak element in the paper relates to the role of circRELL1 in restricting HSV-1 productive infections. The effects observed are modest at best and the biological impact appears limited. It is also entirely unclear how circRELL1 might act to restrict HSV-1. The paper would significantly benefit from the authors extending their analysis to include 2-3 additional circRNAs that are commonly regulated by herpesviruses to determine (i) whether similar effects are observed and (iii) to determine whether a compound effect can be achieved by targeted silencing of multiple IFN-responsive circRNAs at the same time.

Minor comments

- Line 55: To this reviewer's knowledge, only eight routinely infect humans (HSV1, HSV2, VZV, HCMV, HHV6, HHV7, EBV, and KSHV). It's possible the authors are considering HHV6A and HHV6B as distinct but this is not really the case.

- Line 57: As written it implies that therapeutic agents capable of clearing VZV exist which is not really the case.
- Line 132: The authors inclusion of previously published data (e.g. HCMV) along with reams of their own makes for a compelling analysis.
- Line 582: A number of the sequencing datasets are not yet publicly available. This should be rectified before publication.

2. Significance:

Significance (Required)

It is clear that a lot of work has gone into this manuscript and there a reams of data that will provide a great resource for mining to the wider herpesvirus community. While this naturally leads to a more descriptive nature, it does not undermine the value of the data. However, as indicated below, more functional validation is required if this work is to provide a significant step forward in our understanding of how and why IFN-induced circRNAs might be important for combating viral infections.

3. How much time do you estimate the authors will need to complete the suggested revisions:

Estimated time to Complete Revisions (Required)

(Decision Recommendation)

Between 1 and 3 months

Yes

Review #2

1. Evidence, reproducibility and clarity:

Evidence, reproducibility and clarity (Required)

Summary:

In this manuscript, Dremel et al explore the interplay between herpesvirus and circRNAs. This team has been a pioneer in the field and has made important discoveries about the regulation of circRNAs during herpesviral infection. While past work focused on the gamma-herpesviruses, here in this study, they expand their work to alpha and beta HV as well as extending their findings into animal models. They found consistent increase in CircRNA levels and found that these circRNAs are mostly resistant to viral-induced RNA decay. They then uncover that these circRNA can be induced by interferon which indicates that circRNAs could act as a first line of anti-viral defense. Consistently, they found that circRELL1, previously identified as a circRNA inhibiting KSHV lytic infection, can also restrict HSV-1. This seemingly conserved anti-viral capacity could point to an evolutionarily conserved mechanism of defense against infection. This study combines together an impressive amount of sequencing data and elegantly draws parallels between the various HV. While this is a complex story, bringing together expression levels of mRNA, circRNA and even diving into miRNA networks, this is extremely well written and as a reader, you feel transported into a well-crafted journey that keeps uncovering novel and exciting findings.

Major comments:

- there is a back and forth between the HVs that are included in the study: fig 1 has HSV, CMV and KSHV; while fig 2 is HSV, KSHV and MHV68. For clarity and consistency, the authors should consider including all 4 representative HV in their figures.
- this might be a misunderstanding that just needs clarification: CIRCscores provide a measure of CircRNA reads over their linear counterparts: during host shutoff, the linear counterpart would likely decrease and therefore the CIRCscore would artificially go up. So the fact that the CIRCscore remains constant over lytic reactivation in KSHV and MHV68, wouldn't that indicate that instead the CircRNA level go down?
- what happens to circRNAs in HCMV infection as they do not encode an endoRNase?
- line 238: the authors should discuss why CpG and poly I:C treatment largely failed to induce expression of CircRELL1
- line 289: if CircRELL1 is induced in response to infection, it could be interesting to induce its expression after infection (maybe a DOX-inducible promoter on the lentivirus) instead of prior to infection (which might hinder infection and hide more significant effects later).
- line 329: the authors mention that the mRNA targets of SOX carry a "degenerate motif": do the circRNA downregulated during KSHV infection contain such motif?

Minor comments:

- figure 3B should show p-values

- figure 3B: showing expression levels of the endoRNases could provide some context for the extent of their effect
- I would refrain from using sentences referring to phenotype "trending upward" which basically reflects that the results are not significant in either upward or downward direction.

2. Significance:

Significance (Required)

CircRNAs are only beginning to emerge as important regulators of gene expression in cells. Only recent developments in sequencing technology have allowed scientists to even detect the presence of these small RNA. Their roles and mechanism of induction remain largely uncharacterized, let alone in the context of viral infection. This team has pioneered the exploration of CircRNA in KSHV and is now poised to extend their findings to other members of the herpesvirus family. This study will be of interest to a broad audience, both virologist looking to better understand the viral-host battle during infection and RNA biologists seeking to better characterize how gene expression can be controlled with circRNA. There is also major therapeutic potential with these types of approaches.

3. How much time do you estimate the authors will need to complete the suggested revisions:

Estimated time to Complete Revisions (Required)

(Decision Recommendation)

Less than 1 month

No

Review #3

1. Evidence, reproducibility and clarity:

Evidence, reproducibility and clarity (Required)

The authors characterized circRNAs that can escape degradation in cells infected by alpha-, beta and gamma herpesviruses both in cultured cells and in mice. During lytic infection, ten circRNAs were modulated across the virus subfamilies, and 67 circRNAs were upregulated after virus infection or treatment with interferon- β or interferon- γ . The authors examined in detail interferon-induced circRNA circRELL1 and noted that this circRNA suppresses lytic infection, likely by interacting with the mTOR pathway and promoting cell proliferation. They also made the astonishing observation that the circRNAs were more resistant to cleavage by virus-encoded nucleases than their linear counterparts. This is a comprehensive study that reveals circRNA key players that control lytic and latent herpesvirus infections.

****Comments:****

1. Linear mRNA abundances are decreased after infection, but linear-derived circRNA abundances are upregulated. What determines increased circRNA abundances when their linear counterparts become limiting? Is the rate of splicing/back-splicing altered in infected cells? Is nuclear-cytoplasmic transport of circRNAs changed?
2. CircRELL1 loss of function experiment: Does the employed siRNA affect the abundance of linear RELL1 mRNA?
3. CircRELL1 gain of function experiment: How was this performed?
4. Why are circRNA and, especially circRELL1, resistant to degradation in interferon-treated cells when OAS genes, and presumably RNaseL, are upregulated?

2. Significance:

Significance (Required)

This study enhances our knowledge of circRNAs in viral infections.

3. How much time do you estimate the authors will need to complete the suggested revisions:

Estimated time to Complete Revisions (Required)

(Decision Recommendation)

Less than 1 month

No

Full Revision

Manuscript number: RC-2023-02175

Corresponding author(s): Joseph Ziegelbauer

[Please use this template only if the submitted manuscript should be considered by the affiliate journal as a full revision in response to the points raised by the reviewers.]

*If you wish to submit a preliminary revision with a revision plan, please use our "Revision Plan" template. **It is important to use the appropriate template to clearly inform the editors of your intentions.**]*

1. General Statements [optional]

This section is optional. Insert here any general statements you wish to make about the goal of the study or about the reviews.

Enclosed please find the revised manuscript entitled, "Interferon-induced circular RNAs escape herpesvirus host shutoff and suppress lytic infection," for your consideration. We believe this study is appropriate for publication given the conceptual advances and potential broad interest as human circular RNAs affecting herpesvirus infections is an expanding field.

We and others have discovered various circular RNAs are made by human viruses. Here we extend this work across the herpesvirus family, to span alpha-, beta-, and gamma-herpesviruses. Our approach identified a cohort of host circular RNAs which respond to diverse herpesvirus infections. We included an interactive tool with a simple interface for readers to query expression changes of their protein-coding genes and circular RNAs of interest in various infection models. We also examined the mechanism underlying circular RNA response to infection and found a number of circular RNA which are induced by type I and II interferons. During lytic herpesvirus infection, viral endonucleases degrade host RNA. Unexpectedly, we found circular RNAs, as a population, are resistant to this phenomenon. Circular RNAs were unaffected by ectopic viral endonuclease expression, unlike their colinear gene products. We propose a two-pronged model in which interferon-stimulated genes may encode both mRNA and circular RNA with antiviral activity, such as in the case of circRELL1. This finding was recently published in *PNAS*, where we described the mechanism of circRELL1 restriction of gamma-herpesvirus infection—a finding now extended to HSV-1 in our new manuscript. This is critical in cases of host shutoff, such as alpha- and gamma-herpesvirus infection, where the mRNA transcripts are degraded, but circular RNAs escape degradation.

We suggest that this manuscript may be of interest to alpha-, beta-, and gamma-herpes virologists and those studying host-virus interactions. Additionally, RNA biologists studying non-coding RNAs could enjoy our studies and the future experimental plans that they might initiate.

We thank the reviewers that have helped improve our manuscript. We also note that the reviewers were positive and did not dispute major conclusions of this manuscript.

This section is mandatory. Please insert a point-by-point reply describing the revisions that were already carried out and included in the transferred manuscript.

REVIEWER COMMENTS

Reviewer #1

Summary

The manuscript by Dremel et al reports on IFN-induced circRNAs that are produced in response to herpesviral infections and that are resistant to virus-mediated degradation (e.g. host shut-off). They provide a wealth of interesting data on circRNA production changes in response to infection with alpha, beta, and gamma herpesviruses and identify a number of host circRNAs that are commonly regulated across all subfamilies. Using circRELL1 as an example, they demonstrate a modest impact on productive HSV-1 infections (which follows up on their previous report of circRELL1 impacting KSHV lytic infection). They subsequently propose a new model in which a subset of IFN-stimulated genes produce both mRNAs and circRNAs with the potential for antiviral activity, the latter operating as a 'workaround' for avoiding virus-mediated shutoff.

Evidence, reproducibility and clarity

Major Comments:

- Line 146: The authors extended their analysis to HSV-1 latently infected mouse trigeminal ganglia but potentially missed a trick but not including an analysis of HSV-1/VZV infected human trigeminal ganglia for which potentially compatible datasets are available (PMID: 29563516).
Response: All differential expression analysis in this study was performed for infected samples relative to a paired uninfected sample. While we appreciate the suggestion, the suggested dataset lacks a paired uninfected sample it is not compatible with our profiling method in comparing host circRNAs that change in expression after infection.
- Line 264: The authors note that a direct relationship in gene expression changes was observed between some circRNAs and mRNAs and not others. However, a gene level analysis may be problematic here as many genes encode multiple distinct transcript isoforms that are variably regulated e.g. during infection. A transcript level analysis (e.g. using Kallisto/Salmon) might enable the authors to specifically link circRNAs with individual mRNA isoforms which would be valuable information in the context of interferon-driven gene expression. Alternatively, the lack of a direct relationship in gene expression changes may also link to variability in circRNA decay / half-lives. The authors could potentially solve this using a metabolic labelling approach to measure mRNA and circRNA decay rates for a subset of the genes of interest.
Response: We have performed transcript level analysis using the Salmon tool for our bulk RNA-Seq data analyzed in Fig. 4. Expression changes, including all gene reads, circRNA reads, and transcript isoform reads are graphed in Supplementary Fig. 4-4. Our major point, namely that select circRNAs including EPST11, SP100, B2M, ZCCHC2, WARS1 reside within colinear genes that are upregulated to a similar degree after interferon stimulation (Supplementary Fig. 4-4A) was echoed by the new transcript level analysis (Supplementary Fig. 4-4B).
- Line 273: The weak element in the paper relates to the role of circRELL1 in restricting HSV-1 productive infections. The effects observed are modest at best and the biological impact appears limited. It is also entirely unclear how circRELL1 might act to restrict HSV-1. The paper would significantly benefit from the

authors extending their analysis to include 2-3 additional circRNAs that are commonly regulated by herpesviruses to determine (i) whether similar effects are observed and (iii) to determine whether a compound effect can be achieved by targeted silencing of multiple IFN-responsive circRNAs at the same time.

Response: In this study, we propose a model in which interferon stimulated genes may encode both mRNA and circRNA species, of which only the circRNA escapes virus mediated RNA decay. All points of this model were demonstrated for circRELL1, including its ability to impair lytic replication for KSHV (Tagawa *et al.* 2023 *PNAS*) and HSV-1 (Fig. 5). We agree with the reviewer's comment and are very interested in extending our analysis to additional circRNAs and potential compound effects; however we feel performing functional studies for additional circRNA species is beyond the scope of the current study.

Minor comments:

- Line 55: To this reviewer's knowledge, only eight routinely infect humans (HSV1, HSV2, VZV, HCMV, HHV6, HHV7, EBV, and KSHV). It's possible the authors are considering HHV6A and HHV6B as distinct but this is not really the case.

Response: 'The Family Herpesviridae: a brief introduction' chapter of Fields Virology 7th edition states, "Nine herpesviruses have been identified that have humans as their primary host: herpes simplex viruses 1 and 2 (HSV-1 and HSV-2), human cytomegalovirus (HCMV), varicella-zoster virus (VZV), Epstein-Barr virus (EBV), human herpesviruses 6A, 6B, and 7 (HHV-6A, HHV-6B, and HHV-7), and Kaposi sarcoma herpesvirus (KSHV, also known as Kaposi's sarcoma-associated herpesvirus and HHV-8)." The International Committee on Taxonomy of Viruses (<https://ictv.global/taxonomy>) recognized HHV-6A and HHV-6B as distinct viruses in 2012. As summarized by Ablashi *et al.* *Archives of Virology* 2014 (24193951), this decision was based on decades of data recognizing that HHV-6A and HHV-6B are genetically and biologically distinct species. Intravariant recombinants are not found; they are distinguished by their epidemiology and disease associations and biological differences in protein function and tissue tropism.

- Line 57: As written it implies that therapeutic agents capable of clearing VZV exist which is not really the case.

Response: We have updated the text (lines 57-59) to read, "To date only varicella zoster virus has an FDA-approved vaccine. Additionally, we lack antivirals capable of targeting the latent reservoir and there is no therapeutic agent capable of clearing these viruses."

- Line 132: The authors inclusion of previously published data (e.g. HCMV) along with reams of their own makes for a compelling analysis.

Response: We're glad you appreciated the analysis.

- Line 582: A number of the sequencing datasets are not yet publicly available. This should be rectified before publication.

Response: All datasets are currently uploaded to SRA and will be publicly released before publication.

Significance

It is clear that a lot of work has gone into this manuscript and there are reams of data that will provide a great resource for mining to the wider herpesvirus community. While this naturally leads to a more descriptive nature, it does not undermine the value of the data. However, as indicated below, more functional validation is

required if this work is to provide a significant step forward in our understanding of how and why IFN-induced circRNAs might be important for combating viral infections.

Reviewer #2

Summary

In this manuscript, Dremel et al explore the interplay between herpesvirus and circRNAs. This team has been a pioneer in the field and has made important discoveries about the regulation of circRNAs during herpesviral infection. While past work focused on the gamma-herpesviruses, here in this study, they expand their work to alpha and beta HV as well as extending their findings into animal models. They found consistent increase in CircRNA levels and found that these circRNAs are mostly resistant to viral-induced RNA decay. They then uncover that these circRNA can be induced by interferon which indicates that circRNAs could act as a first line of anti-viral defense. Consistently, they found that circRELL1, previously identified as a circRNA inhibiting KSHV lytic infection, can also restrict HSV-1. This seemingly conserved anti-viral capacity could point to an evolutionarily conserved mechanism of defense against infection. This study combines together an impressive amount of sequencing data and elegantly draws parallels between the various HV. While this is a complex story, bringing together expression levels of mRNA, circRNA and even diving into miRNA networks, this is extremely well written and as a reader, you feel transported into a well-crafted journey that keeps uncovering novel and exciting findings.

Evidence, reproducibility and clarity

Major comments:

- There is a back and forth between the HVs that are included in the study: fig 1 has HSV, CMV and KSHV; while fig 2 is HSV, KSHV and MHV68. For clarity and consistency, the authors should consider including all 4 representative HV in their figures.

Response: We recognize presenting data from multiple viruses and human and mouse models may cause some confusion for readers. We appreciate the suggestion, but for the following reasons have chosen to keep Figure 1 and 2 in their current state. Figure 1 and paired supplementary figures was split by species (human, mouse) as the purpose was comparative analysis between viruses and comparisons must be performed in their respective species. In Figure 2, as we discuss in response to Reviewer #2's third comment, excludes HCMV. This was due to technical issues, namely the absence of ERCC spike-in controls. We feel that presenting the HCMV data with our other virus models in Figure 2 would be potentially misleading to readers.

- This might be a misunderstanding that just needs clarification: CIRCscores provide a measure of CircRNA reads over their linear counterparts: during host shutoff, the linear counterpart would likely decrease and therefore the CIRCscore would artificially go up. So the fact that the CIRCscore remains constant over lytic reactivation in KSHV and MHV68, wouldn't that indicate that instead the CircRNA level go down?

Response: We are not sure what data the reviewer is referring to regarding "CIRCscore remains constant over lytic reactivation in KSHV and MHV68". There are no plots including CIRCscore for KSHV lytic reactivation data. The results in Figure 1C containing CIRCscore are from de novo infection of LECs with KSHV and MHV68 analysis is not included in Figure 1. Supplementary Figure 1-2 contains CIRCscore for MHV68 lytic reactivation and here we observe a general increase in CIRCscore with reactivation—this occurs in tandem with increasing circRNA levels. Additionally, we performed transcript specific analysis in Fig. 3A for all genes which give rise to circRNAs in our models, and our findings demonstrate that circRNA species are more resistant to virus-induced downregulation than their colinear mRNA.

Full Revision

- What happens to circRNAs in HCMV infection as they do not encode an endoRNase?
Response: We performed similar analysis for HCMV infection as that included in Figure 2:

This was performed with the intention of it serving as a “negative control” regarding what happens to host transcripts during lytic infection and in the absence of a viral endoRNase. However, we still observed a global decrease in host transcripts by 72 hours post infection. Additionally, classic housekeeping genes such as *GAPDH*, *HPR1*, *RPS13*, and *7SL* were downregulated. We are concerned this finding is artifactual as the HCMV RNA-Seq was the only model we did not sequence ourselves and lacked ERCC spike-in controls. We suspect that the total amount of RNA in HCMV infected cells increases at late times, thus comparison of host transcripts in uninfected samples and late lytic time points results in a relative down sampling. We are not an HCMV lab and lack the ability to confirm our theory. In the absence of ERCC spike-in controls we cannot account for global changes in total RNA content and as such chose not to include HCMV global profiling as it may mislead readers. This technical issue (undersampling at late time points) would not impact our findings in Figure 1 and suggests that the number of upregulated circRNAs is larger than we report.

- line 238: the authors should discuss why CpG and poly I:C treatment largely failed to induce expression of CircRELL1
Response: Additional text has been added to the discussion (lines 391-403) and reads, “Of the stimuli tested, circRELL1 expression was largely unaffected by poly I:C and CpG treatment (Sup. Fig. 4-2). Poly I:C and CpG DNA are recognized by endosomal toll-like receptors (TLR), TLR3 and TLR9, whereas LPS is sensed by TLR4 on the cell surface. The subcellular localization of TLRs and relative amplitude of their downstream signaling may impact circRELL1 upregulation. Additionally, poly I:C treatment can induce robust RNase L-dependent circRNA decay (40); a phenomenon that may disguise poly I:C dependent circRELL1 upregulation.”
- line 289: if CircRELL1 is induced in response to infection, it could be interesting to induce its expression after infection (maybe a DOX-inducible promoter on the lentivirus) instead of prior to infection (which might hinder infection and hide more significant effects later).
Response: We agree that lentiviral transduction prior to infection may cause off-target effects and for this reason all comparisons were made relative to a control which was transduced with a lentivirus expressing circGFP. Due to the rapid pace of HSV-1 infection (DNA replication begins ~3 hpi, infectious viral progeny detectable after ~6 hpi), lentiviral transduction after HSV-1 infection is not technically feasible.
- line 329: the authors mention that the mRNA targets of SOX carry a "degenerate motif": do the circRNA downregulated during KSHV infection contain such motif?

Response: We have performed motif scanning for up and downregulated circRNAs present in our KSHV lytic reactivation model. These findings are now present as Supplementary Table 2 with text discussing them in the results (lines 190-199) and discussion section (lines 360-363).

Results section text: “We examined if the tri-modal distribution of host circRNAs during KSHV reactivation may be explained by sequence differences. A degenerate motif “UGAAG” can increase substrate recognition for the KSHV endoRNase, SOX (54). We posited that underrepresentation of this motif in circRNAs may allow them to escape KSHV host shut off. We performed motif scanning (Sup. Table 2) for our tri-modal circRNA populations to assess incidence of the motif. Proportions of motif-harboring circRNAs did not correlate with up- or down-regulation upon reactivation. Overall density of degenerate motifs was, however, highest in downregulated circRNAs and lowest in upregulated circRNAs, suggesting a potential role of UGAAG frequency in SOX resistance.”

Discussion section text: “Motif scanning suggested high frequency of the consensus “UGAAG” may be partially responsible for host circRNA up- or down-regulation during KSHV lytic reactivation (Sup. Table 2). This, however, does not rule out the possibility that SOX targeting may be ameliorated by additional factors.”

Supplementary Table 2. Analysis of SOX substrate-targeting motif

The presence of SOX substrate-targeting motifs “UGAAG” was determined for subsets of human circRNAs from our KSHV reactivation model (iSLK-BAC16).

Subset: Log ₂ FC (3dpi/Uninduced)	# circRNA	# circRNA in circAtlas	# UGAAG motifs	# Motifs per circRNA	# Motifs per 1000 nt	UGAAG+ circRNA	
						# circRNA	% Total
Upregulated: ≥0.5	42	39 circRNA (22459 nt)	71	1.9	3.2	37	88%
Unchanged: <0.5 and >-0.5	28	28 circRNA (20,975 nt)	95	3.7	4.5	26	93%
Downregulated: ≤-0.5	30	25 circRNA (12,900 nt)	104	4.7	8.1	22	73%

Minor comments:

- figure 3B should show p-values

Response: We have now updated Figure 3B to include p-values.

- figure 3B: showing expression levels of the endoRNases could provide some context for the extent of their effect

Response: We measured transcript expression using qPCR and also included histograms for Thy1.1 surface expression from our flow cytometry analysis before and after Thy1.1 enrichment, see Supplementary Figure 3-2.

- I would refrain from using sentences referring to phenotype "trending upward" which basically reflects that the results are not significant in either upward or downward direction.

Response: We have removed any text referring to statistically insignificant trends in our circRELL1 functional studies (lines 304-307).

Significance

CircRNAs are only beginning to emerge as important regulators of gene expression in cells. Only recent developments in sequencing technology have allowed scientists to even detect the presence of these small RNA. Their roles and mechanism of induction remain largely uncharacterized, let alone in the context of viral infection. This team has pioneered the exploration of CircRNA in KSHV and is now poised to extend their findings to other members of the herpesvirus family. This study will be of interest to a broad audience, both virologist looking to better understand the viral-host battle during infection and RNA biologists seeking to better characterize how gene expression can be controlled with circRNA. There is also major therapeutic potential with these types of approaches.

Reviewer #3

Summary

The authors characterized circRNAs that can escape degradation in cells infected by alpha-, beta and gamma herpesviruses both in cultured cells and in mice. During lytic infection, ten circRNAs were modulated across the virus subfamilies, and 67 circRNAs were upregulated after virus infection or treatment with interferon- β or interferon- γ . The authors examined in detail interferon-induced circRNA circRELL1 and noted that this circRNA suppresses lytic infection, likely by interacting with the mTOR pathway and promoting cell proliferation. They also made the astonishing observation that the circRNAs were more resistant to cleavage by virus-encoded nucleases than their linear counterparts. This is a comprehensive study that reveals circRNA key players that control lytic and latent herpesvirus infections.

Evidence, reproducibility and clarity

- Linear mRNA abundances are decreased after infection, but linear-derived circRNA abundances are upregulated. What determines increased circRNA abundances when their linear counterparts become limiting? Is the rate of splicing/back-splicing altered in infected cells? Is nuclear-cytoplasmic transport of circRNAs changed?

Response: Our analysis of transcriptional activity for a subset of host circRNAs upregulated during HSV-1 infection found a decrease in Pol II occupancy on gene promoters and decrease in nascent (4sU-labeled) RNA-Seq reads (Supplementary Fig. 3-1). This data suggests circRNA expression changes for this subset are related to co-transcriptional processes such as increased rates of back-splicing, see lines 202-216. We have previously performed a similar analysis using 4sU-Seq in a KSHV lytic reactivation model (Tagawa et al. 2023 *PNAS*). The analysis also supported co-transcriptional circRNA upregulation, e.g. increased back-splicing rates.

- CircRELL1 loss of function experiment: Does the employed siRNA affect the abundance of linear RELL1 mRNA?

Response: By 12 hours post HSV-1 infection, *RELL1* mRNA is undetectable by qPCR—likely due to host shut off. To ensure our siRNAs were specifically targeting circRELL1 we collected RNA at 48 hours post siRNA treatment, in the absence of infection, and measured transcript levels. We observed significant depletion of circRELL1 while the colinear mRNA was unaffected, this data is now present as Supplementary Figure 5-1 and shown here.

- CircRELL1 gain of function experiment: How was this performed?

Response: Thank you for catching this, we apologize for omitting details regarding how circRELL1 gain and loss of function experiments were performed. Text describing our approach is now contained in the methods section (lines 576-597).

- Why are circRNA and, especially circRELL1, resistant to degradation in interferon-treated cells when OAS genes, and presumably RNaseL, are upregulated?
Response: RNase L is activated during an innate immune response by the interferon stimulated genes, 2'-5'-oligoadenylate synthetases (OASs). OASs must bind dsRNA to synthesize 2-5-oligoadenylate, which then activates RNase L. RNase L is an endoribonuclease which when activated targets cytoplasmic RNA for decay. Prior work from Chu-Xiao Liu *et al.* (2019 *Cell*), analyzed RNase L-dependent circRNA depletion following immune stimulation. Their study found that immune stimulation via IFN-beta or IFN-gamma treatment did not result in widespread circRNA depletion. Uniquely, poly I:C treatment resulted in RNase L-dependent circRNA decay. Our findings corroborate this study, namely IFN treatment does not result in widespread circRNA depletion. Why RNase L is not activated despite gene expression of the OAS genes may be potentially explained by the absence of dsRNA species and subsequent lack of 2-5-oligoadenylate synthesis.

Significance

This study enhances our knowledge of circRNAs in viral infections.

Dear Dr. Ziegelbauer,

Thank you for the transfer of your revised manuscript from Review Commons to our editorial offices. I have now received the reports from the three referees that were asked to re-evaluate your study, you will find below. As you will see, the referees fully support publication of your study in EMBO reports.

Before I can proceed with formal acceptance, the manuscript now needs formatting according to our journal style. Please carefully review the instructions that follow below.

When submitting your final revised manuscript, we will require:

1) a .docx formatted version of the final manuscript text (including legends for main figures, EV figures and tables), but without the figures included. Figure legends should be compiled at the end of the manuscript text.

2) individual production quality figure files as .eps, .tif, .jpg (one file per figure), of main figures and EV figures. Please upload these as separate, individual files upon re-submission.

The Expanded View format, which will be displayed in the main HTML of the paper in a collapsible format, has replaced the Supplementary information. You can submit up to 5 images as Expanded View. Please follow the nomenclature Figure EV1, Figure EV2 etc. The figure legend for these should be included in the main manuscript document file in a section called Expanded View Figure Legends after the main Figure Legends section. Additional Supplementary material should be supplied as a single pdf file labeled Appendix. The Appendix should have page numbers and needs to include a table of content on the first page (with page numbers) and legends for all content. Please follow the nomenclature Appendix Figure Sx, Appendix Table Sx etc. throughout the text, and also label the figures and tables according to this nomenclature. In this case, I think it will be possible to combine some of the Supplementary Figures to have in the end 5 EV figures. It is not necessary to have one EV figure related to one main figure. Just make sure that the panels are called out correctly.

3) a complete author checklist, which you can download from our author guidelines (<https://www.embopress.org/page/journal/14693178/authorguide>). Please insert page numbers in the checklist to indicate where the requested information can be found in the manuscript. The completed author checklist will also be part of the RPF.

4) that primary datasets produced in this study (e.g. RNA-seq, ChIP-seq, structural and array data) are deposited in an appropriate public database. If no primary datasets have been deposited, please also state this in a dedicated section (e.g. 'No primary datasets have been generated and deposited'), see below.

The accession numbers and database should be listed in a formal "Data Availability" section (placed after Materials & Methods) that follows the model below. This is now mandatory (like the COI statement). Please note that the Data Availability Section is restricted to new primary data that are part of this study. This section is mandatory. As indicated above, if no primary datasets have been deposited, please state this in this section

Data availability

5) We now request the publication of original source data with the aim of making primary data more accessible and transparent to the reader. Our source data coordinator will contact you to discuss which figure panels we would need source data for and will also provide you with helpful tips on how to upload and organize the files.

6) Our journal encourages inclusion of *data citations in the reference list* to directly cite datasets that were re-used and obtained from public databases. Data citations in the article text are distinct from normal bibliographical citations and should directly link to the database records from which the data can be accessed. In the main text, data citations are formatted as follows: "Data ref: Smith et al, 2001" or "Data ref: NCBI Sequence Read Archive PRJNA342805, 2017". In the Reference list, data citations must be labeled with "[DATASET]". A data reference must provide the database name, accession number/identifiers and a resolvable link to the landing page from which the data can be accessed at the end of the reference. Further instructions are available at: <http://www.embopress.org/page/journal/14693178/authorguide#referencesformat>

7) Regarding data quantification and statistics, please make sure that the number "n" for how many independent experiments were performed, their nature (biological versus technical replicates), the bars and error bars (e.g. SEM, SD) and the test used to calculate p-values is indicated in the respective figure legends (also for potential EV and Appendix figures). Please also check that all the p-values are explained in the legend, and that these fit to those shown in the figure. Please provide statistical testing where applicable. Please avoid the phrase 'independent experiment', but clearly state if these were biological or technical replicates. Please also indicate (e.g. with n.s.) if testing was performed, but the differences are not significant. In case n=2, please show the data as separate datapoints without error bars and statistics. See also: <http://www.embopress.org/page/journal/14693178/authorguide#statisticalanalysis>

8) Please also note our reference format:

9) We updated our journal's competing interests policy in January 2022 and request authors to consider both actual and perceived competing interests. Please review the policy <https://www.embopress.org/competing-interests> and update your competing interests if necessary. Please name this section 'Disclosure and Competing Interests Statement' and put it after the Acknowledgements section.

10) We now use CRediT to specify the contributions of each author in the journal submission system. CRediT replaces the author contribution section. Please use the free text box to provide more detailed descriptions and do NOT add an author contributions section to the manuscript text file. See also guide to authors:

<https://www.embopress.org/page/journal/14693178/authorguide#authorshipguidelines>

11) Please add up to 5 keywords to the manuscript and order the manuscript sections like this, using these names:

Title page - Abstract - Keywords - Introduction - Results - Discussion - Materials and Methods - Data availability section - Acknowledgements - Disclosure and Competing Interests Statement - References - Figure legends - Expanded View Figure legends - Tables

12) It seems Table S1 is a dataset, too. Please upload all 7 datasets as original excel files with a legend and a title on the first TAB. Please name these files Dataset EV1 - 7 and change the callouts accordingly using these names.

13) Please enter all the funding information also into our submission system during resubmission and make sure this is complete and similar to the one mentioned in the acknowledgements section of the manuscript text file.

14) Please provide a final abstract with not more than 175 words.

In addition, I would need from you:

- a short, two-sentence summary of the manuscript (not more than 35 words).
- three to four short (!) one sentence bullet points highlighting the key findings of your study.
- a schematic summary figure (synopsis image) in jpeg or tiff format with the exact width of 550 pixels and a height of not more than 400 pixels that can be used as a visual synopsis on our website.

I look forward to seeing a revised version of your manuscript when it is ready. Please let me know if you have questions or comments regarding the revision.

Best,

Referee #1:

I would like to thank the authors for addressing all of my point of concerns - they have significantly improved their manuscript and thoroughly and carefully addressed the reviewers comments.

Referee #2:

The authors have performed admirably in addressing my previous critiques in an appropriate manner and I think the work as presented is excellent.

Referee #3:

The authors have adequately addressed my raised concerns.

Rev_Com_number: RC-2023-02175

New_manu_number: EMBOR-2023-58459V1

Corr_author: Ziegelbauer

Title: Interferon induced circRNAs escape herpesvirus host shutoff and suppress lytic infection

The authors addressed the minor editorial issues.

Dear Dr. Ziegelbauer

Thank you for the submission of your revised manuscript to our editorial offices. We are nearly done. Before we can proceed with formal acceptance, I have these editorial requests I ask you to address in a final revised manuscript:

- Please provide the abstract written in present tense throughout.

- It seems Figs. 5 and 6 have landscape format. Please change this and format the figures as described in our guide for figure preparation:

- Please make sure that the number "n" for how many independent experiments were performed, their nature (biological versus technical replicates), the bars and error bars (e.g. SEM, SD) and the test used to calculate p-values is indicated in the respective figure legends (main, EV and Appendix figures). Please also check that all the p-values are explained in the legend, and that these fit to those shown in the figure. Please provide statistical testing where applicable. Please avoid the phrase 'independent experiment', but clearly state if these were biological or technical replicates. Please also indicate (e.g. with n.s.) if testing was performed, but the differences are not significant. In case n=2, please show the data as separate datapoints without error bars and statistics. See also:

<http://www.embopress.org/page/journal/14693178/authorguide#statisticalanalysis>

If n<5, please show single datapoints for diagrams. Presently, some diagrams don't have statistics or only partial statistics, or miss the n.s. (e.g. panels in Figs. 2, 5 and EV4 and panels 3B, S4A and S6). Moreover:

- Please provide individual figure legends for panels EV 3a-b

- Please indicate the statistical test used for data analysis in the legends of figures 1b; 4a-d; EV 1b; EV 2b.

- Please note that information related to n is missing in the legends of figures 5c, f, h.

- Please format the figure legends for main, EV and Appendix figures) according to our journal style. See the respective section in our guide to authors (please find the link below). Please separate each panel description by a line break and make sure that the panels are listed in alphabetic order. Moreover, please add to each legend a 'Data Information' section explaining the statistics used or providing information regarding replicates and scales.

- Please make sure that all figure panels and Tables are called out separately and sequentially (main, EV and Appendix items). Presently, it seems a callout for panel 5H is missing. Please check.

- The labels in the legends (in all the excel files) for the 7 datasets need to be corrected: e.g. it should be "Dataset EV1" instead of "EV Dataset 1" (please follow the nomenclature Dataset EVx). Please also check that their callouts are correct.

- Please remove the title page from the Appendix file. It is sufficient to state there 'Appendix for ...', followed by the table of contents.

- Please name the Appendix figures Appendix Figure Sx (the S is missing) and make sure this nomenclature is also used for their callouts.

- Please move all the methods information and related references to the main manuscript text file. Please do not provide methods information in the Appendix.

- Please upload 2 separate excel files with the source data - one file per figure.

- Please place Table 1 and 2 after the main and before the EV figure legends.

- Please add a paragraph titled 'Biosafety' to the methods section gathering all information on where and how biosafety-relevant experiments with viruses were performed and that these were approved, and by whom (institution, government).

- Please remove the reagents and tools table from the main manuscript text file. I have attached templates for that in word or excel format. Please upload the filled in table to the manuscript tracking system as 'Reagent Table' file. Please also adjust any callouts to this table. The example linked below shows how the table will display in the published article and includes examples of the type of information that should be provided for the different categories of reagents and tools. Please list your reagents/tools using the categories provided in the template and do not add additional subheadings to the table. Reagents/tools that do not fit in any of the specific categories can be listed under "Other":

https://www.embopress.org/pb%2Dassets/embo-site/msb_177951_sample_FINAL.pdf

Best,

The authors addressed the remaining editorial issues.

Joseph Ziegelbauer
National Cancer Institute
HIV and AIDS Malignancy Branch
10 Center Dr.
Bethesda 20892
United States

Dear Dr. Ziegelbauer,

I am very pleased to accept your manuscript for publication in the next available issue of EMBO reports. Thank you for your contribution to our journal.

Yours sincerely,
